# Illusory sound texture reveals multi-second statistical completion in auditory scene analysis

Richard McWalter [1,2,3]* & Josh H. McDermott [1,2,3,4]*

Sound sources in the world are experienced as stable even when intermittently obscured, implying perceptual completion mechanisms that "fill in" missing sensory information. We demonstrate a filling-in phenomenon in which the brain extrapolates the statistics of background sounds (textures) over periods of several seconds when they are interrupted by another sound, producing vivid percepts of illusory texture. The effect differs from previously described completion effects in that 1) the extrapolated sound must be defined statistically given the stochastic nature of texture, and 2) the effect lasts much longer, enabling introspection and facilitating assessment of the underlying representation. Illusory texture biases subsequent texture statistic estimates indistinguishably from actual texture, suggesting that it is represented similarly to actual texture. The illusion appears to represent an inference about whether the background is likely to continue during concurrent sounds, providing a stable statistical representation of the ongoing environment despite unstable sensory evidence.

[1] Department of Brain and Cognitive Sciences, MIT, Cambridge, MA 02139, USA. [2] Center for Brains, Minds and Machines, MIT, Cambridge, MA 02139, USA. [3] McGovern Institute for Brain Research, MIT, Cambridge, MA 02139, USA. [4] Program in Speech and Hearing Biosciences and Technology, Harvard University, Cambridge, MA 02138, USA. *email: mcwalter@mit.edu; jhm@MIT.EDU

Perception consists of inferences about the state of the world given sensory stimulation. These inferences are typically ill-posed, akin to solving an equation with multiple unknowns, as when estimating sound sources from a mixture of sources, or three-dimensional depth relationships from two-dimensional images. As such, most perceptual inferences necessitate assumptions about the world variables being estimated—knowledge of environmental regularities that have been internalized over development or evolution. Such inferences and their constraining assumptions can be revealed by illusions—artificial situations or stimuli that are erroneously perceived, illustrating the assumptions that produce correct inferences most of the time in natural environments.

One class of perceptual inferences, and associated illusions, occur when perceptual systems "fill in" missing data, as when objects are occluded or sound sources are masked. Sensory traces of objects and sounds are physically interrupted in such situations, and yet we usually experience them as continuous. Classic examples in vision include illusory contours[1], often termed "modal" completion because the contours are subjectively visible (Fig. 1a, left), and "amodal" completion[2], when an object is seen to continue beneath an occluding surface (Fig. 1a, right). In audition, perceptual completion is known to occur for speech and tone-like sounds[3–12], which are heard as continuous even when brief segments are removed and replaced with noise, provided that the noise is sufficiently intense as to plausibly mask the speech or tone (Fig. 1b). Because the stimuli in such "continuity illusions" are equally consistent with discontinuous sound sources or objects, they appear to represent an inference that the sound source was likely to have continued during the noise. However, previously documented effects are for the most part short-lasting, spanning brief gaps of a few hundred milliseconds, and have mostly been limited to sounds that might be produced by individual sound sources in the world.

Auditory scenes frequently contain sound textures—sounds produced by the superposition of many sound sources, such as a room of people talking or clapping, swarms of insects, or rain[13,14]. Sound textures often continue over long periods of time, providing a background to transient events within an auditory scene. Given that the background is often physically continuous but intermittently obscured by other sound sources (Fig. 1c, d), we investigated whether textures might be inferred to continue during such interruptions. Texture completion seemed potentially interesting in part because textures are believed to be represented with relatively low-order statistics that summarize acoustic information over time[14–21], raising the question of whether statistical representations could be extrapolated over time.

Here we describe a class of illusions in which texture subjectively continues when followed by sufficiently intense masking noise, producing vivid illusory percepts (Fig. 1e). Examples of the illusion can be found here: http://mcdermottlab.mit.edu/textcont.html.

Unlike the brief illusory continuity heard for speech, tones or other non-stationary sounds, illusory textures persist for up to several seconds. Because textures are temporally dense, stochastic, and in many cases defined only by relatively simple summary statistics, the perceptual completion appears to be mediated by extrapolated statistics. The extended duration of the illusory sound facilitated investigation of the underlying representation. Illusory texture appears to be incorporated into the statistical estimation process for texture, biasing the perception of subsequent textures in the same manner as physically realized texture. This effect suggests that the underlying perceptual completion processes instantiate representations like those produced by actual texture. The results reveal a form of perceptual completion that is statistical in nature, extends over periods of seconds, and that appears to be specific to dense stationary sounds, suggesting a difference in the representation of textures and discrete events.

## Results

**Overview of experiments**. We conducted a series of experiments to document the illusion, establish the conditions in which it occurs, and probe its representational basis. We first set out to

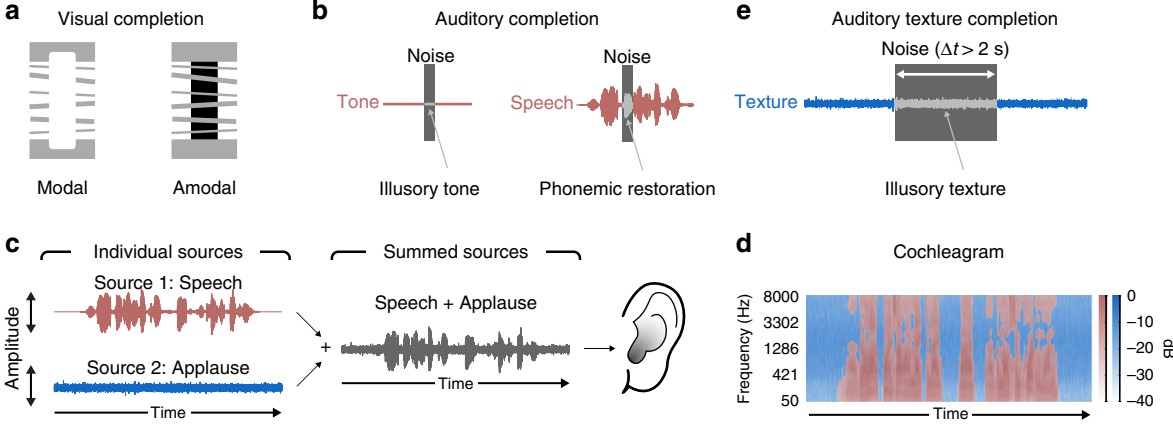

**Fig. 1** Perceptual completion effects. **a** Examples of "modal" and "amodal" visual completion. In the modal case (left), we see a white rectangle overlaid on gray bars. However, the rectangle edges do not produce contrast where they overlap the background. In the amodal case (right), the black rectangle is seen to continue behind the gray bars. **b** Traditional examples of auditory completion. Tones can be heard to continue when interrupted with a sound (e.g., noise) sufficiently loud as to plausibly mask the tone. The tone is heard to continue even though it is not physically present during the noise. Illusory phonemes (or phonemic restoration) can also occur under similar conditions, when a short segment of speech is replaced with a louder sound. Here and in e, the gray shading of the sound waveform symbolizes illusory sound that is heard despite not being present in the stimulus. **c** Example auditory scene in which two sound sources (speech and applause) combine to form a single waveform that arrives at the ear. **d** Masking of background sound texture. The speech sound has more energy during certain time-frequency windows, intermittently masking the background applause sound. The red areas show windows where the speech has more energy than the applause. The blue areas show windows where the applause has more sound energy than the speech. **e** Illusory texture from auditory texture completion. Illusory texture can be heard when a texture is interrupted with another sound sufficiently loud as to mask the texture. The texture is heard to continue even though it is not physically present during the interrupting sound. Texture completion is differentiated from other forms of auditory completion in the long temporal extent over which illusory textures can be heard (often exceeding 2 s)

validate our informal observations of illusory texture. We asked listeners to judge the continuity of a variety of sounds interrupted by noise, exploring whether the effect was specific to textures (Experiment 1). We then measured the temporal extent of the illusion (Experiments 2 and 3). Next, we characterized the relationship of the illusion to masking (Experiment 4) and physical continuity (Experiment 5). We then used the temporal integration of texture to ask if illusory texture is represented like actual texture (Experiment 6). Finally, we explored whether illusory texture could also occur when textures were part of an auditory scene with other sounds (Experiment 7).

**Illusory texture occurs for most stationary sounds**. As an initial validation of our subjective observations, we asked listeners to judge the continuity of a variety sounds interrupted by noise. We selected a set of 80 'inducer' sounds that included some textures as well as a wide assortment of non-stationary sounds, including speech and music along with non-stationary environmental sounds (see Supplementary Table 1 for list). In a first experiment (Experiment 1a), a 2 s segment of the inducer sound was replaced with white masking noise (Fig. 2a). We asked listeners to judge whether the sound was continuous during the noise. Listeners were not told that the sound would be consistently absent during the noise. To validate these judgments we also included trials where the sounds were unambiguously physically present or absent during the noise (Fig. 2b, left). We ran one version of the experiment in normal laboratory conditions, and another using online participants via Amazon's Mechanical Turk service. The online experiment enabled the collection of data from large numbers of participants in order to better resolve differences between large numbers of stimuli. The control conditions in which the inducer sounds were unambiguously present or absent during the noise produced high and low reports of subjective continuity, respectively, suggesting that online listeners were correctly performing the task (Fig. 2b, right).

Despite the fact that the inducer sounds were never physically present in the main condition of interest, some sounds were nearly always judged to continue through the noise (Fig. 2c). However, sounds varied in the extent to which they elicited illusory continuity: some sounds were almost never heard to continue. The results were similar between in-lab and online conditions (Fig. 2c; $r = 0.92$, $p < 0.001$; Pearson correlation), but the online data-enabled highly reliable results across sounds, making it clear that the differences between sounds are robust and replicable (Fig. 2d; split-half reliabilities were $r_{MTurk} = 0.95$, $p < 0.001$ and $r_{Inlab} = 0.81$, $p < 0.001$ for the two versions of the experiment).

To assess whether the phenomenon was specific to textures, which are distinguished by the stability of their statistics over time (i.e., stationarity), we examined whether the variation in continuity could be predicted by a sound's temporal stationarity. We quantified stationarity with the standard deviation of a sound's texture statistics measured in successive excerpts (Fig. 2e; we computed the measure for excerpt lengths ranging from 125 ms to 2 s, and averaged the results)[20,22]. Sounds that were more stationary (sound textures) tended to be heard as continuous during the interrupting noise, whereas sounds that were less stationary (e.g., speech/event sounds) did not (Fig. 2f). Continuity was predicted fairly well by this measure of estimated stationarity ($r = 0.84$, $p < 0.001$; Pearson correlation; Fig. 2g).

Because the masker used in the experiment was white noise, which itself is highly stationary, the results could in principle have been driven by the statistical similarity between the inducer sound and the masker (which is greater for more stationary inducer sounds; Supplementary Fig. 1) rather than by the stationarity of the inducer per se. We note that the masker in this experiment was constrained to be able to continuously mask each of the 80 inducer sounds, preventing us from using highly non-stationary sounds as the masker (because they generally contain spectrotemporal gaps through which other signals can be glimpsed). However, it is possible to deviate from white noise to some extent. To assess the importance of masker-inducer similarity, we conducted an additional experiment (Experiment 1b) in which half the trials featured masking noise synthesized from the average statistics of the set of 80 inducer sounds (using texture synthesis; Fig. 2h, Supplementary Fig. 1a, b). This "mean noise" masker was less stationary than white noise, and the statistical similarity between inducer and masker was dissociated from the inducer stationarity (Supplementary Fig. 1e).

Although there was some effect of the masker sound on reported continuity (listeners were more likely to report hearing the inducer sound as present with the mean noise masker than the white noise masker; Supplementary Fig. 1c), inducer stationarity remained strongly predictive of perceived continuity ($r = 0.78$, $p < 0.001$; Pearson correlation, with the mean noise masker; Fig. 1h, Supplementary Fig. 1f). By contrast, statistical similarity to the mean noise masker was only weakly correlated with the illusion ($r = -0.25$, $p = 0.03$; Pearson correlation; Supplementary Fig. 1g). This result provides support for the notion that the stationarity of the inducer is critical for multi-second illusory continuity.

To further distinguish stationarity from similarity to white noise we separately examined the effect of uniformity across time from that of uniformity across frequency. To obtain comparable measures in the two dimensions we simply measured the variability of the cochleagram across time (averaged over frequency) or across frequency (averaged over time), yielding measures of temporal and spectral "density" for each inducer sound (Supplementary Fig. 2). Unsurprisingly, the temporal density of the inducer was correlated with our stationarity measure ($r = 0.89$) and predictive of illusory continuity ($r = 0.76$, $p < 0.001$; Pearson correlation; data from Experiment 1a). By contrast, the spectral density was marginally negatively correlated with illusory continuity ($r = -0.22$, $p = 0.05$; Pearson correlation; data from Experiment 1a). This analysis provides further evidence that stationarity, rather than similarity to noise, is critical to the illusion.

**Multi-second illusory continuity is specific to textures**. The results of Experiment 1 suggest that multi-second perceptual completion could be specific to texture. However, given that the statistical structure of textures is highly predictable from one second to the next, it also seemed plausible that the extent of completion could instead be determined by the extent to which the perceptually important properties of a sound were predictable. To more thoroughly document the temporal extent of perceptual completion for different sounds, we conducted two follow-up experiments.

In Experiment 2, we varied the duration of segments of white masking noise from short (125 ms) to long (2 s) and measured illusory continuity for a range of inducer sounds (Fig. 3a). Stimuli were 6 s long. The first 2 s was always a segment of an inducer sound, while the last 4 s interleaved 2 s of noise and 2 s of the inducer. For the short masker durations, the noise occurred as a series of pulses interspersed with the inducer. For the longest masker duration, there was a single segment of noise followed by the inducer. Listeners again reported whether they heard a sound as continuous or discontinuous during the interrupting noise. Given the large number of conditions, this experiment was conducted online (using Amazon MechanicalTurk) to obtain reliable results.

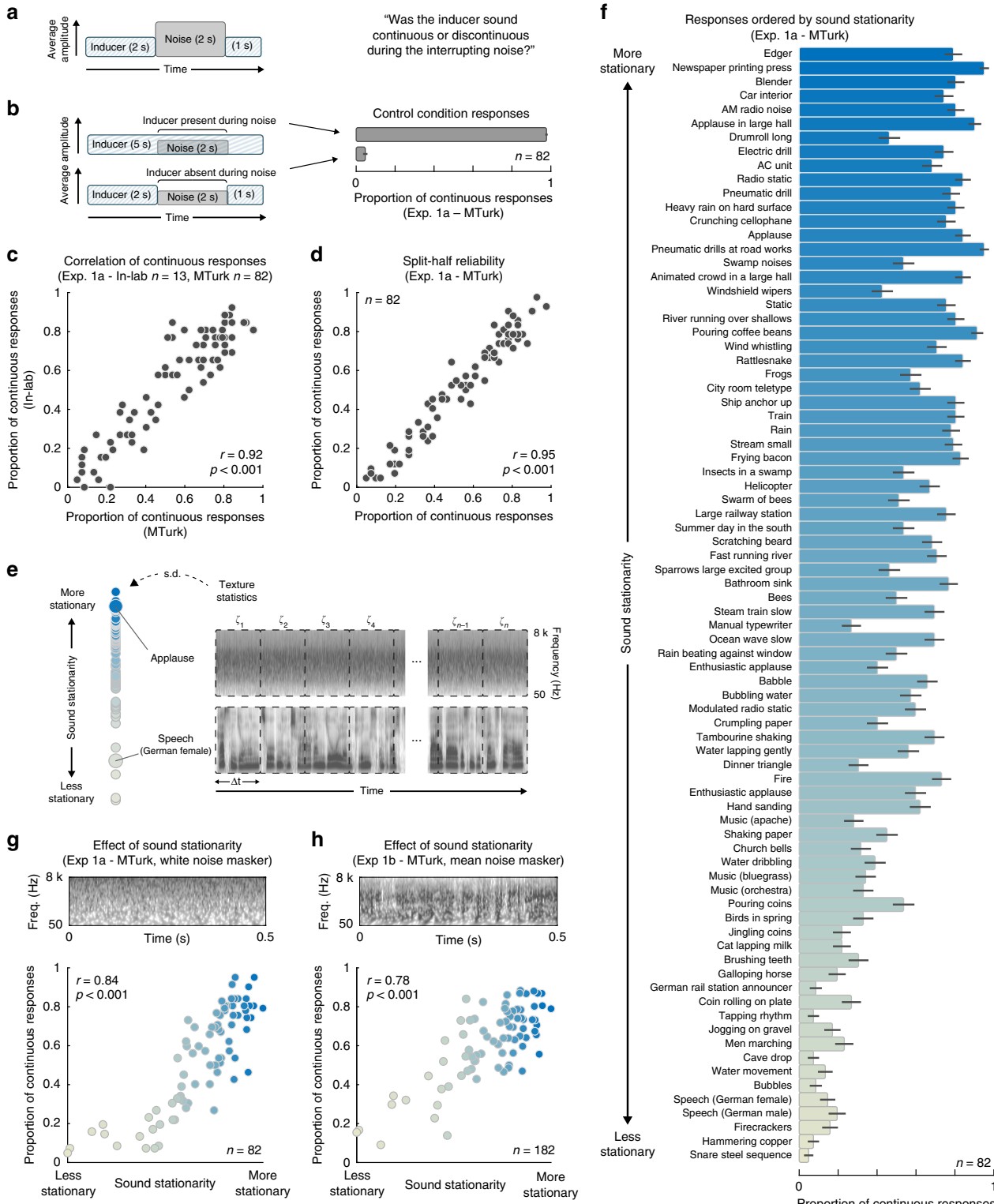

To better distinguish between stationarity and predictability, we included four classes of sounds: (1) stationary environmental sounds (sound textures), (2) non-stationary environmental sounds (waves, crumpling paper, church bells etc.), (3) speech and music, and (4) sounds with periodic modulations at rates ranging from 2 to 8 Hz (a galloping horse, sawing wood, ticking clock etc.). The salient properties of sounds in the latter condition were predictable over time scales of several seconds, but the sounds lacked the stable statistics of sound textures (because the

modulations were slow relative to the windows over which stationarity was measured, such that different windows tended to yield different statistics). As shown in Fig. 3b, these four groups of sounds were differentiated as desired via quantitative metrics of stationarity and periodicity.

Figure 3c shows the average results for the four sets of sounds. Consistent with previous work on phonemic restoration and tone continuity, all sound types exhibited considerable perceptual continuity for the shortest masker durations (no significant

**Fig. 2** Experiment 1—illusory continuity across natural sounds. **a** Schematic of stimulus and task. On each trial, one of 80 real-world sound recordings was interrupted with masking noise. Listeners reported whether the inducer sound continued during the noise. Here, and in other figures, height of inducer and noise segments in stimulus schematic symbolizes the relative sound level (in this case with the noise higher in level than the inducer). **b** Control conditions. Top left: inducer was physically and audibly present during the interrupting noise. Bottom left: inducer was physically and audibly absent during the noise. In both cases, the noise was 6 dB lower in level (quieter) than the inducer, such that the inducer was detectably present during the noise in the top condition, and unambiguously absent during the noise in the bottom condition. Right: results of control conditions of Experiment 1a (online). **c** Reliability of results across in-lab and online experiments. Each dot plots the results for one of the 80 sounds (the proportion of participants who judged the sound to continue during the noise). **d** Reliability of online results across participant splits. Plot shows the split-half correlation for the online experiment for one example split. The Pearson correlation was averaged over 10,000 random splits of participants. **e** Stationarity measure (negative logarithm of standard deviation of statistics measured in different sound segments). Segment durations varied from $\Delta t = 125$ ms to $\Delta t = 2$ s. **f** Results of Experiment 1a (proportion of trials on which the inducer was judged to continue during the noise). Sounds are sorted by stationarity (the value of which is denoted by the bar color). **g** Mean proportion of continuous responses from Experiment 1a (with white noise masker) plotted vs. sound stationarity (also denoted by dot color, to aid comparison to **f**). Top: excerpt of white noise masker signal, displayed as a cochleagram. **h** Mean proportion of continuous responses from the mean noise condition of Experiment 1b plotted vs. sound stationarity. Conventions are as in (**g**). Top: excerpt of mean noise masker signal, displayed as a cochleagram. The mean noise is visibly less stationary than the white noise. Error bars plot SEM

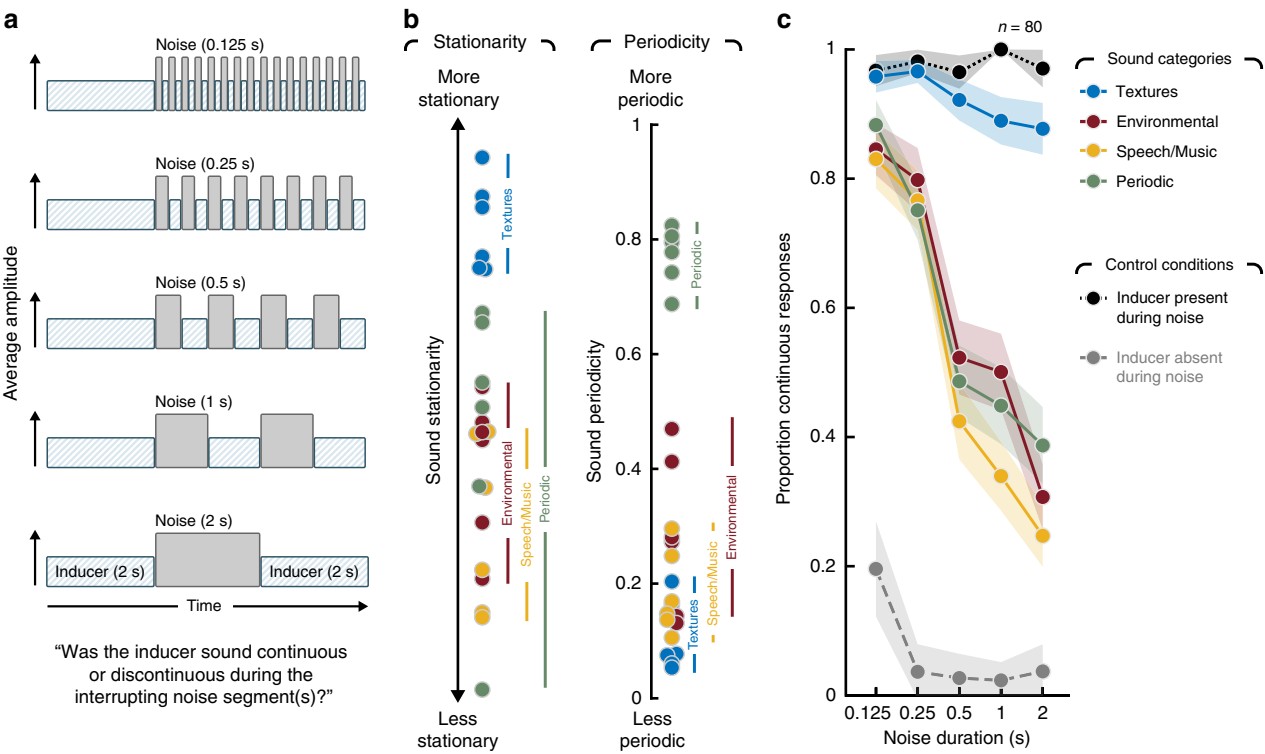

**Fig. 3** Experiment 2—illusory continuity across masker durations. **a** Schematic of stimulus conditions in Experiment 2. **b** Stationarity and periodicity of sounds used in the experiment. Each data point plots the stationarity or periodicity of an individual real-world sound recording used as an inducer sound in Experiment 2 (dot color denotes the four sound categories). **c** Results for Experiment 2. Control conditions show results when the inducer was physically present or absent during the interrupting noise segment(s) (same as control conditions of Experiment 1; see Fig. 2b for schematics). Shaded region plots SEM of individual data points

variation across sound class for the shortest duration, F(3, 237) = 1.52, $p = 0.21$; repeated measures ANOVA (rmANOVA)). But whereas sound textures exhibited a high degree of illusory continuity for the longer masker durations as well (no significant variation in perceived continuity of textures across masker durations, F(4, 316) = 1.94, $p = 0.10$; rmANOVA), the other three sets of sounds did not. In all three cases, the proportion of continuous responses declined sharply with increasing masker duration (Environmental sounds: F(4, 316) = 27.41, $p < 0.001$; Speech/Music: F(4, 316) = 32.67, $p < 0.001$; Periodic sounds: F(4, 316) = 19.13, $p < 0.001$; rmANOVA), producing an interaction between the effect of duration and sound class (F(12, 948) = 6.55, $p < .001$, rmANOVA). In particular, there was a noticeable drop in perceptual continuity between 250 and 500 ms for the

non-textures (Environmental Sounds: t(79) = 4.82, $p < 0.001$; Speech/Music: t(79) = 4.84, $p < 0.001$; Periodic sounds: t(79) = 2.68, $p = 0.0091$; two-tailed paired t-test), suggestive of a limit on the extent to which non-textures can be perceptually filled in.

The results suggest that sound textures are inferred to continue over a wide range of masker durations, but that other types of sounds do so only across shorter extents. The results also indicate that the extended perceptual completion for textures is not merely the consequence of their predictability (see Supplementary Fig. 3 for an analysis of the results of Experiment 1 in terms of periodicity, which provides further support for this conclusion).

As an alternative measure of the extent of illusory continuity, in Experiment 3 we asked listeners to indicate the duration over which they heard continuity, using an analogue response

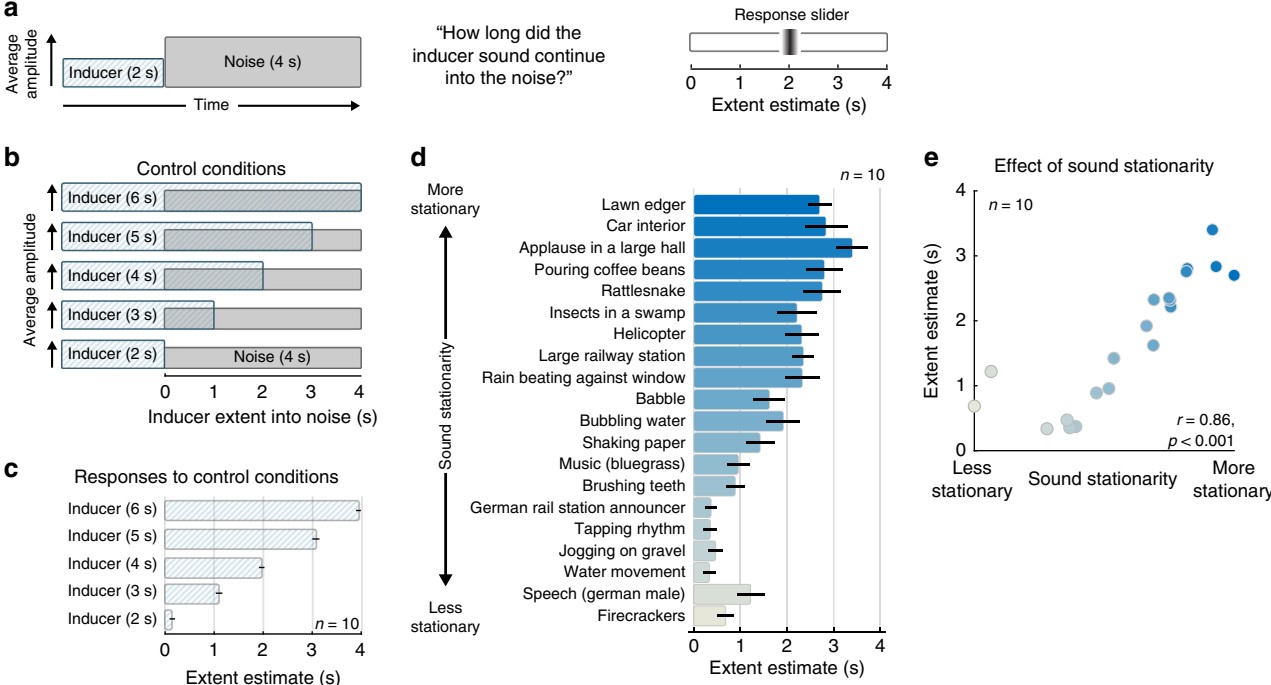

**Fig. 4** Experiment 3—estimation of the extent of illusory continuity. **a** Schematic of stimulus and task for main condition of Experiment 3. Inducer sounds used in the experiment were real-world sound recordings. Following the noise, participants adjusted a slider to indicate the extent over which they heard the inducer sound to continue into the noise. **b** Schematic of stimuli for control conditions, in which the inducer was higher in level than the noise and persisted for different amounts of time after the noise onset. **c** Mean estimated persistence for control conditions, pooled across inducer sounds. **d** Mean estimated extent of persistence for main condition of Experiment 3, plotted separately for different inducer sounds. Sounds are sorted by their stationarity. Bar color denotes inducer sound stationarity. **e** Mean extent estimate plotted vs. inducer sound stationarity (also denoted by the dot color, to aid comparison to panel d). Error bars plot SEM

measure. Listeners heard 6 s of audio (Fig. 4a). The first 2 s was an original sound recording—either a texture, or one of a small selection of non-stationary sounds. The last 4 s consisted of white noise that could plausibly mask the original sound. Listeners reported the extent to which the original sound continued into the noise by adjusting a slider on a graphical interface once the sound had finished. The slider covered the range from 2 s (when the noise started) to 6 s (when the noise ended). To confirm that participants could accurately perform this task we included control trials in which the texture was higher in level relative to the noise and physically continued either 0, 1, 2, 3, or 4 s into the noise (Fig. 4b). As shown in Fig. 4c, listeners positioned the slider accurately for inducer sounds that physically extended into the noise.

In the main experimental trials (where the inducer sound was lower in level than the noise, and physically absent during the noise), the mean slider setting generally exceeded 2 s for texture sounds (Fig. 4d, top of results graph), though it never reached all the way to 4 s. By contrast, the mean settings for non-stationary sounds (Fig. 4d, bottom of graph) were largely under 1 s. The sounds thus significantly varied in the temporal extent of perceived continuity (F(1,9) = 90.02, p < 0.001; rmANOVA), with the extent of continuity being correlated with stationarity (Fig. 4e, r = 0.86, p < 0.001; Pearson correlation). The results suggest that there are limits to the extent of illusory continuity for textures, but provide additional evidence that they can be heard to continue substantially longer than non-textures.

**Illusion is heard only when the texture is plausibly masked.** We next tested whether illusory texture could be explained as the inference of a masked source. We first measured whether the

illusion depended on whether the interrupting noise was high enough in level to mask the texture, were they present concurrently.

Listeners performed two tasks with related stimuli (Fig. 5a). In the first task, we measured the detection of a 2 s target texture superimposed on white noise, as a function of signal-to-noise ratio (SNR). On each trial, listeners heard two noise bursts and were asked to identify if the target texture was present in the first or second interval. Trials were grouped into blocks related to a particular target texture. In the second task, listeners heard a 5 s texture interrupted by 2 s of white noise, and reported whether the texture was continuous or discontinuous during the noise. In both experiments, the level of the texture relative to the noise (SNR) was varied from −18 dB to +12 dB. We sought to determine whether illusory continuity would be perceived only at SNRs that would produce masking of the texture were it actually present during the noise.

We conducted separate experiments with real-world recorded textures and synthetic textures generated from their statistics. Because the synthetic textures were generated from statistics, they additionally served to substantiate whether the perceptual completion underlying the illusory texture was statistical in nature.

As expected, detection of the target improved with increasing SNR, for both real-world and synthetic textures (Fig. 5b—red curves, real F(5, 45) = 139.9, p < 0.001, synthetic F(5, 45) = 62.2, p < 0.001; rmANOVA). In the companion continuity task, listeners were more likely to report the inducer texture as continuous during an interrupting noise segment when the noise level was high (Fig. 5b—blue curves, real F(5, 45) = 173.9, p < 0.001, synthetic F(5, 45) = 91.8, p < 0.001; rmANOVA). Results were again similar for real-world and synthetic textures. Comparison of the masking curves (red) and continuity curves (blue) shows that perceived continuity was high only at SNRs for

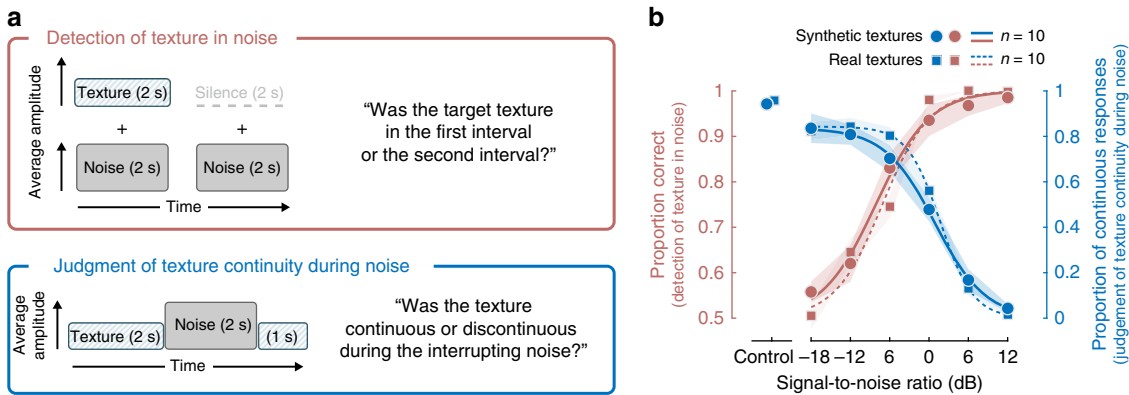

**Fig. 5** Experiments 4a and 4b—relation of masking to illusory texture. **a** Top: Detection of texture in noise (red; Experiment 4a). The stimulus consisted of two Gaussian noise signals that were 2 seconds in duration, separated by a 400 ms inter-stimulus interval. One of the signals had a texture excerpt superimposed on the noise. Listeners identified the stimulus interval that contained the texture. Bottom: Judgment of texture continuity during noise (blue; Experiment 4b). Stimulus construction and task were identical to that of Experiment 1. The stimulus consisted of a 2 s excerpt of a texture, immediately followed by 2 s of noise, immediately followed by another 1 s of the texture. We ran two versions of each experiment: one with real-world texture recordings, and one with synthetic textures generated from statistics measured from real-world recordings. Listeners judged whether the inducer texture was continuous or discontinuous during the interrupting noise segment. The SNR was varied across the same range in the two experiments. **b** Results of Experiments 4a and 4b. In Experiment 4a (red), the detectability of the target texture in noise decreased with SNR. In Experiment 4b (blue), listeners more readily reported the texture as continuous with decreasing SNR. The two lines correspond to the two versions of the experiments (circles and solid lines: synthetic textures; squares and dashed lines: real-world texture recordings). Control conditions featured stimuli where the inducer was higher in level and physically present during the interrupting noise (same as one of the control conditions in Experiment 1; see Fig. 2b, left panel). Shaded regions show SEM of the individual data points

which listeners had difficulty detecting the texture when it was physically present in the noise. The detectability and continuity of a texture were thus negatively correlated across SNRs (synthetic textures: $r = -0.88$, $p = 0.021$; recorded textures: $r = -0.83$, $p = 0.040$; Pearson correlation).

The SNR at which the noise masked the target varied across sounds, presumably related to some acoustic characteristic of the target sound texture (e.g. the extent of amplitude modulation; see Supplementary Fig. 4 for results for individual textures). However, the SNR at which continuity was experienced also varied somewhat across sounds, and the masking thresholds and continuity thresholds were significantly correlated across sounds ($r = 0.75$, $p < 0.001$; Pearson correlation, corrected for attenuation).

**Illusion eliminated by silent gaps between texture and noise.** To further test whether illusory continuity was linked to whether texture could plausibly continue during masking sounds, we measured the effect of short gaps placed between the texture and the masking noise. Listeners were presented with 5 s excerpts of textures with intervening segments of white noise (Fig. 6a). The noise either abutted the texture or was temporally offset by a brief 200 ms gap on either or both sides. In all cases, the noise was substantially higher in level than the texture excerpts, set individually to mask each texture based on a pilot version of Experiment 4 (see Methods). Listeners were asked to report what they heard during the noise by choosing one of six possible response contours indicating the presence of (illusory) texture over time: continuous throughout, fade-out, present only at beginning and end of the noise ('dip'), present only in the middle of the noise ('glimpse'), fade-in, and absent throughout (Fig. 6b). We again conducted separate experiments with real-world recorded textures and synthetic textures generated from their statistics. Here we present the results with synthetic textures; see Supplementary Fig. 5 for similar results with real-world recordings.

To ensure that listeners would accurately report each of these percepts were they to occur, the experiment also included control trials in which the initial 2 s of texture was 6 dB higher in level than the noise (making the texture audible when superimposed

on the noise) and where the texture was modulated in amplitude to follow the contour of one of the six response choices (Fig. 6b). For instance, for the "dip" control, the texture remained physically present for a short period into the noise, then faded out, then physically faded back in prior to the end of the noise. The results for these control conditions indicate that listeners accurately reported each of the six percepts when they were unambiguously present in the stimulus (Fig. 6c; listeners chose the correct response contour well above chance in all cases; t(9) > 8.29, $p < 0.001$ in each condition; two-tailed single sample $t$-test).

As shown in Fig. 6d, when the noise was contiguous with the texture, listeners predominantly reported the texture as continuing throughout the noise (two-tailed paired $t$-tests comparing the proportion of continuous responses to each of the other responses were significant in each case, t(9) > 4.1, $p < 0.01$). However, silent gaps substantially altered the illusory texture. When gaps were inserted both before and after the noise, the percept of texture during the noise was largely eliminated (listeners predominantly reported the texture as being absent throughout; two-tailed paired $t$-tests comparing the proportion of absent responses to each of the other responses were significant in each case, t(9) > 3.51, $p < 0.01$). A gap after the masker did not on its own eliminate illusory texture (the proportion of continuous responses was still far above chance; t(9) = 6.08, $p < 0.001$; two-tailed single sample $t$-test). However, the gap increased the tendency of the illusory texture to fade out before the end of the noise (two-tailed paired $t$-test comparing the proportion of fade-out responses in the contiguous and gap-after-noise conditions, t(9) = 3.97, $p = 0.0032$). By contrast, a gap before the masker most often eliminated the illusory texture percept (two-tailed paired $t$-test comparing the proportion of continuous responses in the contiguous and gap-before-noise conditions, t(9) = 4.98, $p < 0.001$), but illusory texture was heard to fade in more often than when gaps were both before and after the noise (two-tailed paired $t$-test comparing the proportion of fade-in responses in the gap-before-and-after and gap-before-noise conditions, t(9) = 3.40, $p = 0.0078$). The latter effect suggests some degree of "retrospective" filling-in[23–25] driven by the texture occurring after the noise.

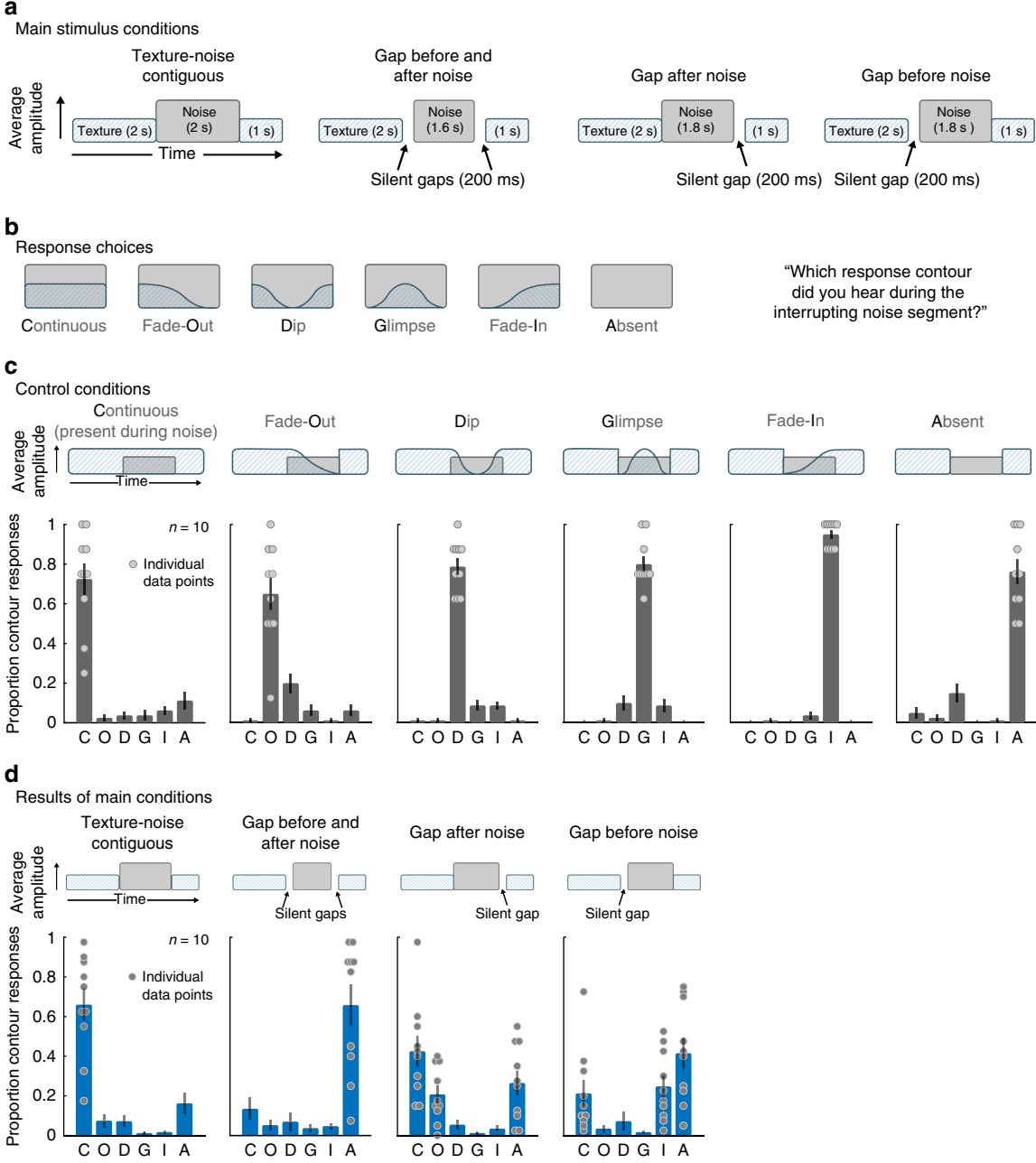

**Fig. 6** Experiment 5—effect of gaps on texture continuity. **a** Listeners heard a synthetic inducer texture interrupted with masking noise and reported their perceptual experience during the interrupting noise segment. The experiment included 4 conditions, differing in the contiguity of the texture and noise (via silent gaps inserted before and/or after the noise). See Supplementary Fig. 5 for analogous experiment with real-world texture recordings (which yielded similar results). **b** Listeners chose one of six response contours to describe their perceptual experience during the interrupting noise segment. The contour response code is indicated as the bolded letter for each response (e.g. "C" for "Continuous"). **c** To confirm task comprehension/compliance, the experiment included control trials where the texture was physically present during the intermediate noise segment and amplitude modulated according to one of the response contours. The stimulus for each condition is schematized above each of the six subplots. Graphs plot the proportion of trials on which each response was chosen. Data for individual participants is plotted as dots for the response choices selected above chance levels for each condition. **d** Results of main experimental conditions. Each subplot corresponds to a condition (shown schematically above). Data for individual participants is plotted as dots for the response choices selected above chance levels for each condition. Error bars show SEM

Together, Experiments 4 and 5 indicate that illusory textures are heard during interrupting noise so long as the texture could have physically continued through the noise. The results suggest that the illusory texture is the result of an unconscious inference about what was likely to be present during the noise. In particular, the experiments indicate that the effect is not simply due to listeners being able to imagine sounds at will

during noise, as brief gaps were sufficient to largely eliminate the effect.

The results of Experiments 4 and 5 also indicate that illusory texture occurs for both for real-world recordings of texture, and for synthetic textures defined only by time-averaged statistics. The latter finding supports the idea that the illusion is the result of extrapolated statistical properties.

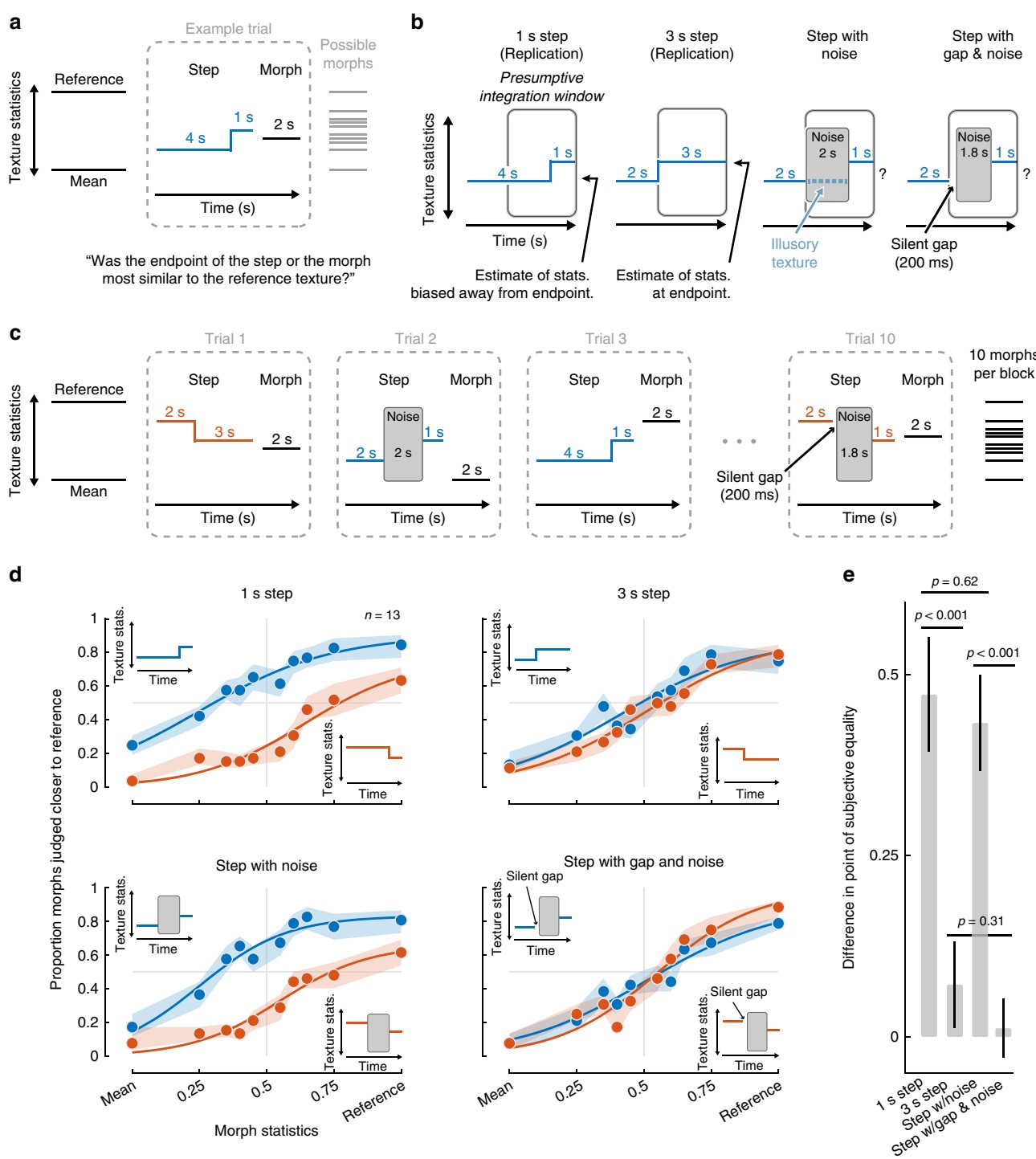

**Illusory texture is represented similarly to actual texture**. What happens in the auditory system to produce illusory texture? To probe the relationship between the representation of illusory texture and that of texture that is physically present in the sound signal, we sought to leverage prior findings that listeners' texture judgments are biased by several seconds of stimulus history[20], presumably reflecting the averaging process underlying texture statistics. If illusory texture is represented like actual texture, for instance with persistent activity in the relevant part of the auditory system, it might similarly bias judgments of subsequent texture.

We modified a texture discrimination experiment from our previous work[20] to incorporate illusory texture. The experiment

(Experiment 6) required listeners to judge which of two sound texture excerpts was most similar to a reference texture. The two excerpts were generated from statistics lying on a line between the mean statistics of a large set of textures and the reference texture. The first excerpt was 5 s and had a subtle change ("step") in statistics at some point during its duration (Fig. 7a). The second excerpt (the "morph") was 2 s and had constant statistics at discrete points sampled on a line between the mean and the reference texture. Listeners were informed that the first excerpt would undergo a change at some point, and they should base their judgments on the sound of the texture at its endpoint. We previously found that these texture steps biased listeners'

**Fig. 7** Experiment 6—statistical integration of illusory texture. **a** Step experiment paradigm. At the start of each trial block, participants were familiarized with a reference texture (synthesized from the statistics of a real-world texture recording) and a mean texture (synthesized from the average statistics of 50 real-world textures). On each trial participants heard two stimuli and judged which was most similar to the reference, basing their judgments on the endpoint of each stimulus. The first stimulus contained a subtle "step" in statistics, the direction and temporal position of which varied from trial to trial. The second stimulus ("morph") had constant statistics in a given trial, but varied between the reference and mean across trials. **b** Main conditions and interpretation. Previous experiments suggested that humans estimate texture statistics using a temporal integration window of several seconds (symbolized in gray). A step occurring within this window should bias judgments away from the endpoint. In the critical condition (third panel), the middle of the step stimulus was replaced with 2 s of Gaussian masking noise. This was expected to elicit an illusory texture during the presumptive integration window that could bias judgments away from the endpoint if the illusory texture replicated the effects of an actual texture in the relevant part of the auditory system. The last condition contained a 200 ms silent gap before the noise. This was a control condition to confirm that eliminating the percept of illusory texture would eliminate any bias observed in the third condition. **c** Trial blocking. The experiment was divided into blocks of 10 trials related to a particular reference texture. The step condition and direction (towards or away from the reference) varied from trial to trial (randomly ordered over the course of the experiment subject to the blocking constraint). The 10 morph positions were presented once per block in random order. **d** Results of Experiment 6. Shaded regions show SEM of individual data points and curves plot logistic function fits. **e** Plot of the bias produced by the step in each condition, quantified as the difference between points of subjective equality for the upward and downward step conditions. Error bars show SEM, obtained by bootstrap (10,000 samples)

judgments away from the endpoint if they occurred within a few seconds of the endpoint, suggestive of an integration window of several seconds for estimating sound texture statistics. We tested whether illusory texture heard during an interrupting noise burst would similarly bias texture judgments.

In the experiment, the first interval always contained a change in statistics, either at 1 s from the endpoint or 3 s from the endpoint (Fig. 7b). Based on our prior work, we expected the 1 s step to bias discrimination judgments, but for the bias to be greatly reduced for the 3 s step, as it occurs outside the apparent integration window for texture[20]. The two critical conditions included interrupting white noise (Fig. 7b). In one condition, the noise extended from 3 s to 1 s from the endpoint. We expected illusory texture to be heard during the noise, extrapolated from the first part of the step. In another condition, the first 200 ms of the noise was replaced by a silent gap, which (based on Experiment 5) we expected to eliminate the illusory texture. The gap also served as a control for the possibility that the noise on its own might bias texture perception. Trials from the different conditions were intermixed within blocks corresponding to a particular reference texture (Fig. 7c), and in all cases listeners were instructed to base their judgments on the end of the step stimulus. If the perception of illusory texture instantiates representations in the auditory system like those of physically realized texture, the step with noise, but not the step with gap and noise, might produce a bias in listeners' judgments.

As shown in Fig. 7d, listeners were biased by texture steps that occurred at 1 s before the endpoint but not at 3 s before the endpoint, as expected. This effect was quantified by fitting psychometric functions to the data and measuring the bias as the difference in the point of subjective equality between the two functions (Fig. 7e). The bias was significantly different for the 1 s step and 3 s step conditions ($p < 0.001$; via bootstrap), and not significantly different from 0 for the 3 s step ($p = 0.22$; via bootstrap). But when the step was interrupted by noise, listeners exhibited a bias comparable to the 1 s step condition, as though the statistics of the initial texture segment were heard during the noise and incorporated into the estimate for the subsequent texture segment (no significant difference in bias between 1 s step and noise conditions, $p = 0.62$; via bootstrap). The bias was eliminated by the silent gap preceding the noise (significantly larger bias for noise than gap condition, $p < 0.001$; no difference between 3 s step and gap conditions, $p = 0.31$; and the bias for the gap condition was not significantly different from 0; $p = 0.77$; via bootstrap).

The results indicate that illusory texture biases texture judgments similarly to texture that is physically present. The effect provides objective evidence for the presence of illusory texture, and places constraints on its representation in the auditory system. In particular, the results suggest that the illusion involves changes to the representations that are integrated to form texture statistic estimates, as these estimates appear to be altered by the illusion.

**Illusory texture is heard within auditory scenes**. In the final experiment (Experiment 7), we probed the real-world relevance of illusory texture by exploring filling-in for typical auditory scenes, which might consist of multiple sound sources that vary in stationarity (Fig. 8a). Listeners were again asked to report whether a sound was continuous or discontinuous during a segment of interfering noise. However, in contrast to Experiments 1–6, two inducers were presented simultaneously to form a simple auditory scene (Fig. 8b). The inducer pairs consisted of one texture and one non-texture sound, selected based on their stationarity (Fig. 8a). On each trial listeners heard the auditory scene (sound pair) interrupted by noise: 2 s of the inducer pair followed by 2 s of interrupting white noise and then another 1 s of the inducer pair. Prior to the auditory scene the listeners were cued with a 2 s excerpt of the target sound whose continuity they were supposed to report. The cued sound could be the texture, the non-texture, or the sound pair (Fig. 8c).

Even though both sounds were present prior to the noise, only the texture sound was heard during the noise to a substantial extent (Fig. 8d; continuity was greater for the texture than for the non-texture, $t(9) = 9.97$, $p < 0.001$, or the scene, $t(9) = 8.09$, $p < 0.001$; two-tailed paired $t$-test). The proportion of trials on which the texture and non-texture were reported to continue was similar to when the sounds were present in isolation (Fig. 8d, replotting data from Experiment 1; no significant difference in continuity for textures in isolation vs. paired with a non-texture: $t(21) = 1.61$, $p = 0.12$; or for non-textures: $t(21) = 0.59$, $p = 0.56$; two-tailed unpaired $t$-test). This result suggests that perceptual completion is determined at the level of individual streams, and that textures are streamed separately from other sounds. It also suggests that the presence of non-stationary sounds does not prevent extended completion of concurrent textures.

## Discussion

We documented an illusion in which sound textures are perceived to continue over periods of up to several seconds when interrupted by a foreground sound, despite being physically absent. This "illusory texture" had a number of diagnostic characteristics. First, the temporal persistence of the illusion appears to be related to the temporal stationarity of the sound. Illusory sound texture lasted up to several seconds during extended masking noise, whereas temporally variable sounds, such as speech, faded out after a few hundred milliseconds. This difference appears to not be simply explained by the timescale over which signals are

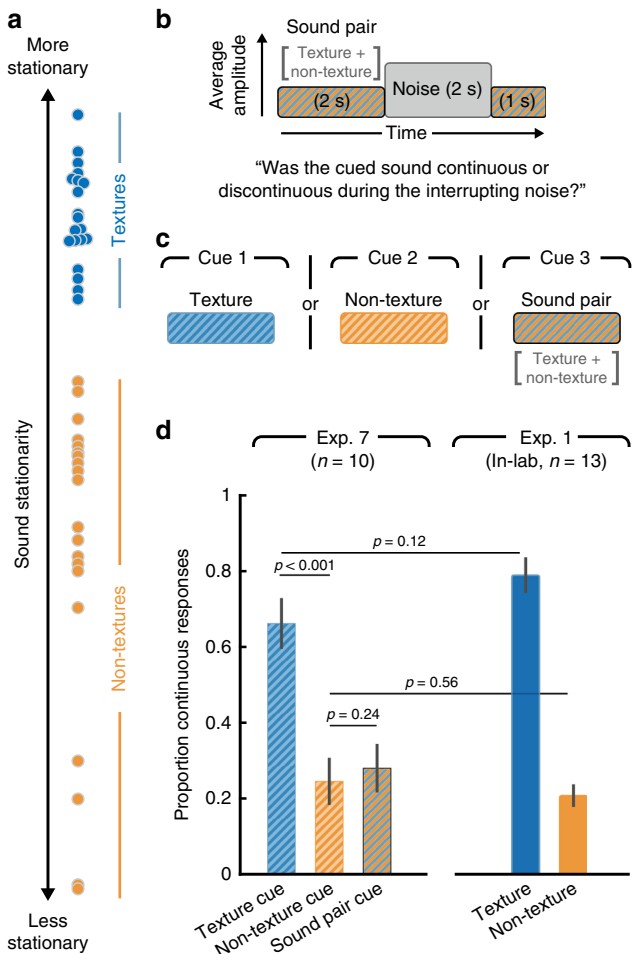

**Fig. 8** Experiment 7—illusory continuity in auditory scenes. **a** Stationarity of the 40 real-world sound recordings used as inducers in Experiment 7. The sounds were divided into two groups: 20 textures and 20 non-textures (selected based on their sound stationarity). **b** Schematic of the main stimulus conditions in Experiment 7. The inducer sound was the superposition of a texture with a non-texture (sound pair), and was interrupted by masking noise as in previous experiments. **c** A cue sound was presented to the listeners prior to each trial, indicating which sound they should be judging the continuity of. The cue could either be the texture, the non-texture or the superposition of the two. **d** Results of Experiment 7, plotting the proportion of continuous responses for each of the three conditions. Right bars: The results of the same task for the 20 textures and 20 non-textures presented in isolation, replotted from Experiment 1. Error bars show SEM

predictable, because temporally sparse periodic sounds (regular rhythms etc., that are highly predictable) also failed to produce multi-second illusory continuity. Extended illusory continuity may thus be unique to statistical representations of sound. Second, illusory texture only occurred when the interrupting foreground sound was temporally contiguous with the texture and sufficiently intense as to plausibly mask the texture. These two results suggest that illusory texture is the result of an inference about whether the texture continues during the masking noise, analogous to that classically thought to occur for tones or speech. Third, leveraging the multi-second averaging process that is believed to underlie texture perception, we found that illusory sound texture was incorporated into subsequent judgments similarly to actual sound texture. This finding provides objective evidence for the illusion and suggests that it could be mediated by persistent activity within the same representational locus as actual

texture. Fourth, when a texture co-occurred with temporally variable sounds, as in typical auditory scenes, the texture was perceived to continue during an interrupting masker even though comparable illusory continuity did not occur for the concurrent nonstationary sounds. This observation suggests that textures are represented as 'streams' within auditory scenes, and that perceptual inferences of continuity are made on individual streams. Given the ubiquity of texture, and of concurrent foreground sounds, the results suggest that the background of auditory scenes is filled in routinely without the listener realizing it, constructing a stable representation of the auditory world from impoverished sensory data.

Perceptual filling-in of speech and tones, and its relation to masking, was first described many decades ago[3–6]. Illusory texture differs from these previously described perceptual completion effects in at least two respects. First, the duration of the effect is much longer. Tones and speech are heard to complete only over a few hundred milliseconds[26–28]. We documented this difference in Experiments 1–3, finding that textures are unusual among natural sounds in persisting for long durations. To our knowledge, the only precedent for this observation is an informal note by Warren[6] that synthetic noise could be heard to continue for relatively long periods of time when interrupted by another masking noise, though this was apparently never substantiated experimentally. The extent of continuity may relate to the likelihood that textures are continuous in the world. Textures are often generated by large numbers of concurrent events (raindrops, hand claps, insect noises), potentially making them unlikely to stop abruptly (unlike, for instance, a person talking). The extent of illusory continuity for different sounds could thus be a rational inference about the likely state of sounds in the world.

It is natural to wonder whether the extent of illusory continuity might alternatively relate to integration windows for texture statistics, which appear to be several seconds in duration[20], on par with the extent of illusory texture. One reason to think that this similarity of time scales is coincidental is that less stationary textures appear to be integrated over longer periods of time, perhaps as needed to obtain stable statistic estimates[20]. By contrast, the extent of illusory continuity is, if anything, longer for more stationary textures (Experiment 3; Fig. 4d). It thus seems more plausible that the extent of perceptual completion reflects priors on the continuity of different types of sound sources.

A second difference with classical continuity effects is that the filling-in of texture is statistical. Unlike tones or speech, textures are stochastic, and on the time scales at which the continuity occurs can only be described and extrapolated in terms of the distribution of their features. There are some previously documented examples of perceptual completion of abstracted properties of sound, in that frequency and amplitude modulation appear to be filled in at a level of representation that discards phase[29,30]. Texture completion appears to be a more extreme version of this sort of phenomenon, in that the content that is completed reflects time-averaged properties of the inducing stimulus. The results underscore the need for models of auditory scene analysis to involve statistical representations.

The statistical nature of illusory texture provides additional evidence for statistical representations of texture. Previous evidence for statistical representations came from the recognizability of textures synthesized from statistics[14,15,18], or from the discrimination of texture excerpts[16]. But for texture that is physically realized in a sound signal, any statistical representation is concurrent with, and presumably derived from, the representation of the acoustic details composing a texture. As a result, the role of statistical representations in perception must be inferred indirectly. Illusory texture, by contrast, occurs without the presence of any underlying acoustic

elements, and when the inducing texture is defined only by statistics, the illusory texture that is heard during the noise must be mediated exclusively by a completed statistical representation. In that sense, it provides the clearest evidence yet that texture perception is based on statistics. We note that the extrapolated statistics could nonetheless instantiate a representation of illusory acoustic details that are consistent with the extrapolated statistics (Experiment 6 is consistent with this possibility).

Experiments with traditional continuity illusions have in some cases suggested explicitly "filled in" neural representations, envisioned as persistent activity from the inducer stimulus continuing during the noise. Evidence for persistent activity has been seen in the primary auditory cortex of cats and monkeys for the original tone continuity illusion[31,32]. And several studies in humans have reported evidence of explicit filling-in during phonemic restoration[33–35]. On the other hand, other studies of tone continuity have suggested more implicit representations of the continuity, with the tone onsets and offsets being suppressed by the noise[36–38], and one report that illusory tones do not produce aftereffects comparable to those induced by real tones[39].

Our results are suggestive of an explicit representation of illusory texture. We found evidence that illusory texture biases the perception of subsequent texture, as though the illusory texture was incorporated into the integration window for estimating texture statistics (Experiment 6). This result is consistent with the idea that perceptual continuity is mediated by persistent activity at the representational stage that is averaged to yield texture statistics. Illusory texture could thus involve a conceptually similar mechanism to that proposed for tones, though in a higher-order representation (e.g., modulation filters[40,41]), and with persistent activity lasting much longer than with tones. But given that the continuity appears to be driven by the statistical properties of a sound, and that these statistics are derived via a multi-second integration process[20], any persistent activity may be driven by feedback from a representation of those statistics, the neural locus of which remains unclear. The strength and duration of the illusion should enable neurophysiological experiments to more definitively probe its neural basis.

Our experiments leave open the role of directed attention in illusory texture. The traditional continuity illusion is believed to be robust to inattention[42], like other aspects of auditory scene analysis that occur on relatively short time scales[43–46]. However, auditory "streaming" phenomena that evolve over seconds are often altered by attention[47,48], raising the possibility that multisecond illusory texture continuity might be similarly affected. The temporal extent of the illusion should enable experiments exploring such effects.

As with other auditory continuity effects, illusory texture is heard in conditions where the texture would be masked were it physically present. Sounds are thus subjectively audible in conditions where they would be undetectable. This subjective audibility is particularly apparent for illusory texture because of how long it lasts, creating vivid percepts that facilitate introspection. The subjective audibility of masked sounds appears to differentiate auditory continuity from perceptual completion effects in vision. The conditions producing auditory continuity (masking) are most analogous to those eliciting "amodal" completion in vision, in which occluded contours (that are obscured by foreground objects) are inferred to extend behind the foreground object (Fig. 1a). Amodal completion has measurable consequences for perception, for instance constraining face recognition[49] and motion integration[50], but the completed contours are not "seen" in the way that perceptually completed sounds are "heard". This difference may reflect the fact that there are two different physical situations to distinguish in vision (occlusion, in which the occluded object cannot possibly be visible, eliciting

amodal completion, and camouflage, in which the foreground object is invisible only because of accidental matches with the background color, eliciting modal completion). Seen contours appear to be a code for occluding objects (as distinct from occluded objects). This distinction is absent in audition, because sounds superimpose rather than occlude. Subjective audibility thus appears to be used to code for the presence of sound sources even in conditions where they are physically undetectable and must be inferred.

## Methods

**Auditory texture model.** The auditory texture model processed an input sound waveform via a cascade of filter banks and measured statistics of their outputs. The filter bank cascade was identical to that described in an earlier paper[20], which was a modification of the original McDermott and Simoncelli model[14]. The first filter bank approximated the frequency selectivity of the cochlea. The envelopes of the output of these filters were then filtered with a temporal modulation filter bank. The "texture" statistics measured from these representations consisted of the mean, coefficient of variation, and skewness of the cochlear envelopes, pair-wise correlations across cochlear envelopes, the power from the modulation filters, and pair-wise correlations across modulation bands. These statistics were identical to those used previously[14,20] except that the cochlear envelope kurtosis was omitted, as we have found it to have little impact on synthesis quality.

For completeness, we describe the model in full here. The rest of this section is reproduced from the methods section of the original paper[20].

To simulate cochlear frequency analysis, sounds were filtered into subbands by convolving the input with a bank of bandpass filters with different center frequencies and bandwidths. We used 4th-order gammatone filters as they closely approximate the tuning properties of human auditory filters and, as a filterbank, can be designed to be paraunitary (allowing perfect signal reconstruction via a paraconjugate filterbank). The filterbank consisted of 34 bandpass filters with center frequencies defined by the equivalent rectangular bandwidth (ERB)$_N$ scale[51] (spanning 50–8097 Hz). The output of the filterbank represents the first processing stage from our model (Supplementary Fig. 6).

The resulting "cochlear" subbands were subsequently processed with a power-law compression (0.3) which models the non-linear behavior of the cochlea[52]. Subband envelopes were then computed from the analytic signal (Hilbert transform), intended to approximate the transduction from the mechanical vibrations of the basilar membrane to the auditory nerve response. Lastly, the subband envelopes were downsampled to 400 Hz prior to the second processing stage.

The final processing stage filtered each cochlear envelope into amplitude modulation rate subbands by convolving each envelope with a second bank of bandpass filters. The *modulation filterbank* consisted of 18 half-octave spaced bandpass filters (0.5–200 Hz) with constant $Q = 2$. The modulation filterbank models the selectivity of the human auditory system and is hypothesized to be a result of thalamic processing[40,53,54]. The modulation bands represent the output of the final processing stage of our auditory texture model.

The model input was a discrete time-domain waveform, $x(t)$. The texture statistics were computed on the cochlear envelope subbands, $x_k(t)$, and the modulation subbands, $b_{k,n}(t)$, where $k$ indexes the cochlear channel and $n$ indexes the modulation channel. The windowing function, $w(t)$, obeyed the constraint that $\sum_t w(t) = 1$.

The envelope statistics include the mean, coefficient of variance, and skewness, and represent the first three marginal moments. The marginal moments capture the sparsity of the time-averaged subband envelopes. The moments were defined as (in ascending order)

$$\mu_k = \sum_t w(t) x_k(t) \tag{1}$$

$$\frac{\sigma_k^2}{\mu_k^2} = \frac{\sum_t w(t) \left( x_k(t) - \mu_k \right)^2}{\mu_k^2} \tag{2}$$

$$\eta_k = \frac{\sum_t w(t) \left( x_k(t) - \mu_k \right)^3}{\sigma_k^3} \tag{3}$$

Pair-wise correlations were computed between the eight nearest cochlear bands. The correlation captures broadband events that would activate cochlear bands simultaneously[14,15]. The measure can be computed as a square of sums or in the more condensed form can be written as

$$c_{jk} = \frac{\sum_t w(t) \left( x_j(t) - \mu_j \right) \left( x_k(t) - \mu_k \right)}{\sigma_j \sigma_k} \tag{4}$$

$$j, k \in [1 \dots 34]$$

such that $(k - j) = [1, 2, 3, 4, 5, 6, 7, 8]$.

To capture the envelope power at different modulation rates, the modulation subband variance normalized by the corresponding total cochlear envelope

variance was measured. The modulation power measure takes the following form

$$\sigma_{k,n} = \frac{\sum_t w(t)\left(b_{k,n}(t) - \mu_{k,n}\right)^2}{\sigma_k^2} \tag{5}$$

$$k \in [1 \dots 34], n \in [1 \dots 18].$$

Lastly, the texture representation included correlations between modulation subbands of distinct cochlear channels. Some sounds feature correlations across many modulation subbands (e.g. fire), whereas others have correlations only between a subset of modulation subbands (ocean waves and wind, for instance, exhibit correlated modulation at slow but not high rates[14]). These correlations are given by

$$c_{jk,n} = \frac{\sum_t w(t)\left(b_{j,n}(t) - \mu_{j,n}\right)\left(b_{k,n}(t) - \mu_{k,n}\right)}{\sigma_{j,n}\sigma_{k,n}} \tag{6}$$

$$j \in [1 \dots 34], (k-j) = [1,2], n \in [3, 5, 7, 9, 11, 13].$$

The texture statistics identified here resulted in a parameter vector, $\zeta$, which was used to generate the synthetic textures.

**Auditory texture synthesis**. Synthetic textures were generated using a previously published variant[20] of the McDermott and Simoncelli[14] synthesis system. The sound texture synthesis system measured the time-averaged statistics from a real-world texture recording using the auditory texture model. The statistics were measured from 7 s sound recording excerpts. The measured statistics were subsequently imposed on the cochlear envelopes of a 5 s Gaussian noise seed via an iterative procedure that adjusted the statistics of the synthetic sound to match the statistics of the target real-world sound using the limited-memory Broyden–Fletcher–Goldfarb–Shanno (L-BFGS) gradient descent optimization algorithm[20]. By seeding the synthesis system with different samples of noise, the procedure allowed for the generation of distinct exemplars that possessed similar texture statistics. Code implementing the synthesis procedure is available online: https://github.com/rmcwalter/STSstep.

**Sound stationarity metric**. To quantify the stationarity of real-world sound recordings, we computed the standard deviation of the texture statistics measured across successive segments of a signal, for a variety of segment lengths (0.125, 0.25, 0.5, 1, and 2 s)[20,22]. For each segment length we carried out the following steps. First, we divided a signal into adjacent segments of the specified duration and measured the statistics in each segment. Second, we computed the standard deviation across the segments. Third, we averaged the standard deviation across statistics within each statistic class (envelope mean, envelope coefficient of variation, envelope skewness, envelope correlations, modulation power, and modulation correlations). Fourth, we normalized the average standard deviation for a class by its median value across sounds (to put the statistic classes on comparable scales, and to compensate for any differences between statistics in intrinsic variability). Fifth, we averaged the normalized values across statistics classes. Sixth, we averaged the resulting measure for the five segment lengths to yield a single measure of variability for each real-world sound recording. Finally, we took the negative logarithm of this quantity, yielding the stationarity measure.

**Statistical distance metric**. To quantify the statistical similarity between the inducer sounds of Experiments 1a and 1b and the noise maskers, we computed the Euclidean distance between the statistics of the noise masker and each inducer sound. Prior to this calculation, individual statistics were z-scored across the set of 81 sounds (the 80 inducer sounds and the masker they were compared to) to put the different types of statistic on comparable scales.

**Spectral and temporal density metrics**. The spectral and temporal sound density was quantified as the negative logarithm of the variance along either axis of the cochleagram of a sound. We computed the cochleagram using the first two stages of the auditory texture model described above. For the temporal density we measured the variance of each frequency channel of the cochleagram across time, averaging over frequency to obtain a single measure. For the spectral density, we measured the variance across frequency at each time point, averaging over time to obtain a single measure. Our analyses used the negative logarithm of these measures.

**Experimental procedure for in-lab experiments**. The Psychophysics Toolbox (Psychtoolbox) for MATLAB was used to play sound stimuli and collect responses. All stimuli were presented at a sampling rate of 48 kHz in a soundproof booth (IAC Acoustics). The presentation sound pressure level (SPL) varied across sounds and experiments, and is specified in the sections below. Sounds were played from the sound card of a MacMini computer over Sennheiser HD280 PRO headphones.

Participants were not screened for inclusion based on whether they heard the illusion or not. In-lab participants were generally compliant with task instructions, as evidenced by good performance on the control trials. The only experiment where participants were excluded was Experiment 6, which involved a subtle

discrimination tasks, and where we excluded participants who could not perform this task above a criterion level. All participants self-reported normal hearing.

All in-lab participants provided informed consent and the Massachusetts Institute of Technology Committee on the Use of Humans as Experimental Subjects (COUHES) approved all experiments.

**Experimental procedure for online experiments**. Experiments 1a, 1b, and 2 necessitated small numbers of trials per participant, and thus required a large number of participants to obtain reliable results. To obtain sufficient numbers we conducted these experiments online using the Amazon Mechanical Turk crowd-sourcing platform. In previous work, we have found that online data can be of comparable quality to in-lab data provided some modest steps are taken to standardize sound presentation[55].

Each participant first adjusted their volume setting so that a calibration Gaussian noise sound was at a comfortable level. The calibration noise root-mean-squared (rms) level was set to the maximum of the rms levels of the experimental stimuli. Participants then completed a 'headphone check' task to help ensure that they were wearing headphones or earphones[56]. Finally, to eliminate participants who were not fully focused on the task, we included practice trials at the beginning of the experiment (described in more detail below). On these trials the inducer stimulus was physically and audibly either present or absent during the noise (because it was higher in level than the noise), such that the stimulus was outside the regime of illusory continuity and there was an unambiguous correct answer. Performance on these trials thus provided a measure of task compliance independent of whether a participant heard the illusion or not. We excluded participants who performed at less than 75% correct on these practice trials. As with the in-lab experiments, participants were not screened for inclusion based on whether they heard the illusion or not.

All online participants provided informed consent and the Massachusetts Institute of Technology Committee on the Use of Humans as Experimental Subjects (COUHES) approved all experiments.

**Masking noise level**. Most experiments used a common stimulus structure in which an inducer sound was interrupted by a white Gaussian noise masker that was higher in level than the inducer. To ensure the interrupting noise would plausibly mask the inducer were they presented simultaneously, we used one of two procedures. For Experiment 1b, we determined the masker level at which the addition of the inducer did not substantially alter the cochleagram of the masker (described in more detail below). For all other experiments, the first author (RM) performed a brief additional experiment to set the level of the masking noise for each inducer sound. In this experiment, 2 s of Gaussian noise was superimposed on 2 s of inducer sound. The 2 s of sound was pulsed on and off, with an inter-stimulus-interval of 400 ms. For each "on" pulse, a new 2 s noise waveform was generated and the 2 s inducer was randomly sampled from a longer 7 s excerpt. The inducer level was fixed at 55 dB SPL. The level of the noise was increased or decreased, in half dB increments starting from 55 dB SPL, until the noise was sufficiently high in level to mask the inducer sound. The resulting signal-to-noise ratio (SNR) between the inducer and the noise is provided for each experiment in Supplementary Tables 1–6.

**Experiments 1a/b overview**. Experiment 1a used a white noise masker and was conducted in-lab and online, to obtain highly reliable mean results (online) while verifying that they would conform to what would be obtained in more controlled listening conditions (in-lab). Experiment 1b used both white noise and "mean noise" maskers, and was conducted exclusively online.

**Experiments 1a/b stimuli**. Eighty natural sounds were selected that spanned a wide range of source types (e.g., speech/music, textures, event-like sounds; see Supplementary Table 1 for a list of the 80 sounds used in Experiment 1) and sound stationarity (Supplementary Fig. 7 shows the measured stationarity of sounds). The stimuli were generated from 5 s excerpts sampled from longer 7 s natural sound recordings (with the excerpt onset randomly sampled between 0 and 2 s). The 5 s excerpt was interrupted with noise, such that the stimulus consisted of the concatenation of 2 s of the real-world sound recording (the "inducer"), 2 s of noise (the "masker"), and the remaining 1 s of the inducer. The inducer and noise were cross-faded over 20 ms using raised cosine ramps applied to the onset and offset of the two signals, to avoid abrupt phase discontinuities in the inducer. As a result, the signal excerpts used to construct the stimulus were 2.01 s, 2.02 s, and 1.01 s in duration, with the midpoint of the crossfade occurring 2 s and 4 s into the resulting stimulus. The intermediate noise segment otherwise replaced the synthetic texture, so that the synthetic texture and the noise were not physically present at the same time.

In Experiment 1a, the masking noise was white Gaussian noise. Experiment 1b additionally used "mean" noise—a synthetic texture generated from the mean texture statistics measured from the 80 inducer sounds, using the texture synthesis procedure described above. We synthesized 5 s excerpts of the noise, from statistics measured from 7 s excerpts of the original inducer recordings. A distinct synthetic noise exemplar was used on each trial of the experiment.

The level of the interrupting noise was set individually for each inducer to mask the inducer were they presented simultaneously. For Experiment 1a, the level was set as described in the previous section (and given in Supplementary Table 1). For Experiment 1b, we used an automated process that measured the difference between the cochleagram of the masker and the superposition of the masker and inducer. The procedure varied the SNR until the measured absolute difference in the cochleagram, averaged across time and frequency, was no greater than 3 dB.

In the online experiment, we presented each inducer sound once per experiment, which for Experiment 1b required that inducer sounds be assigned to a masker condition (white noise or mean noise). Because the stimulus sets needed to be pre-generated for the online testing platform, we generated ten assignments of inducer sounds to masker conditions, randomized subject to the constraint that over the ten stimulus sets, a given inducer sound was paired with each masker condition between 4 and 6 times across the sets. This condition was intended to ensure approximately even sampling of the inducer-masker pairings across participants.

The stimulus construction was otherwise identical across the two experiments.

Control and practice trials in both experiments used stimuli that were identical to those in the experimental trials, except that (a) the inducer was physically present during the intermediate noise segment on half of the trials, and (b) the inducer sound level was set to +6 dB relative to the noise, such that illusory continuity was not heard on the half of the trials where the inducer was physically absent during the noise. These trials were intended to confirm task comprehension (i.e., we expected that participants who were performing the task as desired would judge the trials with the physically present inducer as continuous, and the trials with the physically absent inducer as discontinuous). The control and practice trials used a distinct set of real-world sound recordings.

**Experiments 1a procedure (In-lab)**. Participants judged whether the inducer sound was continuous or discontinuous during the intermediate noise segment. Each participant performed 200 trials: two presentations for each of the 80 sounds and two presentations of the 20 control trials with randomly selected inducer sounds. The presentation level of the inducer sound was set to 55 dB SPL. The presentation order was randomized for each participant. The participants did not receive feedback during the main experiment.

Prior to the main experiment, participants performed 20 practice trials (described above, on which there was an unambiguous correct answer) with feedback. The sounds used for the 20 practice trials did not overlap with the sounds used for the 20 control trials.

**Experiments 1a/b procedure (Online)**. The procedure was identical for Experiments 1a and 1b. Prior to the experiment, participants were instructed to set the volume of their headphones to a comfortable level while listening to a calibration noise signal set to the maximum noise level that would be presented during the main experiment.

Participants then performed a brief headphone check experiment[56] intended to weed out participants who were ignoring the instructions to wear headphones or earphones. They next completed 12 practice trials in which they were asked to report whether they "heard the sound" or "didn't hear the sound" during the interrupting noise segment. As described above, on these trials the inducer sound was higher in level than the masker, and was physically and audibly present or absent during the noise, such that there was an unambiguous correct answer. If the participant's performance met or exceeded 75% correct on the practice section, they continued to the main experiment, otherwise the session was ended. At this point, the calibration noise was presented a second time before the participants began the main experiment to ensure comfortable listening conditions. The main experiment consisted of 92 trials: 80 trials of the main experimental condition (one for each of the 80 sounds) and 12 control trials. The sounds used for the 12 practice trials did not overlap with the sounds used for the 12 control trials.

**Experiments 1a participants**. In-lab: 13 participants completed the experiment (6 female, mean age = 23.5, s.d. = 2.3, 6 self-reported as musicians). Online: 140 participants began the online experiment, of which 58 failed either the headphone check or the practice session. The remaining 82 participants completed the main MTurk experiment (40 female, mean age = 38.8, s.d. = 10.9, 39 self-reported to be musicians). Online participants were unique to Experiment 1a.

**Experiments 1b participants**. 304 participants began the online experiment, of which 122 failed either the headphone check or the practice session. The remaining 182 participants completed the main MTurk experiment (91 female, mean age = 37.0, s.d. = 11.6, 98 self-reported as musicians). Online participants were unique to Experiment 1b.

**Experiment 2 stimuli**. The stimuli consisted of excerpts of real-world sound recordings interrupted with Gaussian noise segments. Twenty-four source recordings were used in the experiment, selected from 4 categories: speech/music, textures, periodic/rhythmic sounds, and environmental sounds (Supplementary Table 2). The textures were selected from a larger set as the most stationary (according to the measure described earlier). To select the periodic sounds, we

measured the normalized auto-correlation of the envelope of the broadband waveform (the Hilbert envelope of the waveform, downsampled to 400 Hz). The height of the largest peak between 125 ms and 500 ms (2–8 Hz) was selected as the measure of periodicity. The six periodic/rhythmic sounds were selected from a larger set as those that had the highest values of this measure. The environmental sounds were selected to be less stationary than the textures and less periodic than the periodic/rhythmic sounds. The speech and music sounds comprised recordings of English, German, bluegrass, and classical.

To generate the stimuli we first selected a 6 s excerpt of one of the sources, where the onset was randomly selected from the first 1 s of a 7 s recording. Segments of the 6 s sample were then replaced with Gaussian white noise. The first 2 s of the sample was left intact. 50% of the remaining 4 s was replaced with noise, such that the stimulus alternated between noise and the inducer. The duration of the noise segments varied logarithmically between 125 ms and 2 s, resulting in between 16 and 1 segments (Fig. 3a). The level of the noise segment(s) was set individually for each inducer to a level that would mask the inducer were the noise and inducer present concurrently (see Supplementary Table 2 for levels). The inducer and noise segments were cross-faded over 20 ms using a raised cosine ramp. The noise segment replaced the real-world sound recording, so that the recorded sound and the noise were not physically present at the same time except during the onset and offset window. There were a total of 120 possible stimuli (24 sounds and 5 masker durations). To avoid priming effects, each participant heard each of the 24 sounds only once, in one of the conditions (see below).

To ensure that listeners understood the task, we also included practice and control trials where the inducer was higher in level than the noise and was either physically present or absent during the noise segment. We chose twenty-four additional sounds spanning the 4 categories described above. The same interrupting noise durations were also used. Stimulus construction was the same except that on the trials where the inducer was physically present, the noise was added to the inducer sound rather than replacing it. The level of the inducer relative to the noise was +6 dB.

**Experiment 2 procedure**. The experiment was conducted online. Participants adjusted their playback level to a comfortable setting using a calibration Gaussian noise stimulus. The level of the calibration noise was set to the maximum noise level that would appear in the main experiment.

Upon passing the headphone check experiment[56], participants completed 18 practice trials with feedback. On each trial, participants reported whether they "heard the sound" or "didn't hear the sound" during the noise. There were a total of 240 possible practice stimuli (24 sounds, 5 noise durations, and with the sound physically present or absent during the noise). The 18 practice trials were randomly selected from this set with the constraint that each sound was only presented once, noise durations were presented either 3 or 4 times, and that there were an equal number of trials where the sound was physically present or absent during the noise.

The task for the main experiment was the same as on the practice trials: participants again heard a 6 s stimulus and reported whether they "heard the sound" or "didn't hear the sound" during the noise. The main experiment consisted of 30 trials: 24 trials of the main experimental conditions and 6 control trials. The main experiment trials were a random subset of the 120 possible trials (24 sounds x 5 masker durations), with the constraint that each participant heard each of the 24 sounds once and each noise duration was presented 4 or 5 times. The sounds on the 6 control trials were distinct from the sounds in the main experimental trials.

**Experiment 2 participants**. 160 participants attempted the experiment. 80 of these passed both the headphone check and the practice portion of the experiment (32 female, mean age = 37.4, s.d. = 12.2, 46 self-reported as musicians). Participants were unique to Experiment 2.

**Experiment 3 stimuli**. We used twenty real-world sound recordings as the inducer sounds (Supplementary Table 3). The stimuli were selected from the larger set used in Experiment 1 to span a range of sound stationarity and source types (e.g., textures, environmental sounds, speech, music and periodic sounds). Stimuli were 6 s in duration and began with a 2 s inducer followed by 4 s of white Gaussian masking noise. The 2 s inducer excerpt was randomly selected from a 7 s recorded sound. The level of the noise was set individually for each inducer to a level that would mask the inducer were the noise and inducer present concurrently (see Supplementary Table 3 for levels).

We included twenty control trials to ensure that participants could perform the task as instructed. The control stimuli consisted of an inducer that was higher in level than the noise and that ended either at the onset of the noise, or 1, 2, 3 or 4 s after the noise onset, continuing into the noise until this point (5 control conditions in total, Fig. 4b). The inducer level was 55 dB SPL and was +6 dB above the noise.

**Experiment 3 procedure**. The experiment was conducted in-lab. Participants estimated the temporal extent by which the inducer perceptually continued into the noise. Following the 6 s stimulus, participants adjusted a slider to indicate the point in time where they heard the inducer end. During the stimulus presentation, participants were shown a progress bar plotting the temporal progression of the

stimulus relative to the slider scale. The slider could be positioned from the noise onset (2-s) to the offset (6-s).

Participants completed 100 trials: 4 presentations of the 20 main experiment conditions (corresponding to the 20 inducer sounds) along with 20 control trials. The pairings of control condition and inducer sound were randomized for each participant, with the constraint that each sound was presented once as a control. The order of trials was randomized for each participant.

Prior to the experiment, participants completed 20 practice trials that had the same structure as the control trials but with different condition/sound pairings. Feedback was provided to the participants after each trial, consisting of a visual indicator on the slider as to when the inducer sound ended.

**Experiment 3 participants**. 10 participants completed the experiment (4 female, mean age = 22.4, s.d. = 2.1, 7 self-reported as musicians). These participants also completed the in-lab version of Experiment 1.

**Experiment 4a/b overview**. Experiment 4a was intended to reveal the level of noise required to mask a sound texture, for comparison to the effect of SNR on perceived continuity measured in Experiment 4b.

**Experiment 4a stimuli**. Twenty sound textures (Supplementary Table 4) were used in the experiment and were selected from a set of sounds used in previous studies of sound texture perception[16]. Each trial presented a 2 s sample of Gaussian noise and a superposition of a 2 s sample of Gaussian noise and a 2 s sound texture excerpt, in random order (Fig. 5a – upper panel). We ran two versions of the experiment: one where the textures were excerpted from real-world recordings, and one where they were excerpted from synthetic textures. For the latter, the stimuli were random excerpts sampled without replacement from twelve 7 s samples synthesized from the statistics of each real-world texture (such that each trial was drawn from a unique exemplar; see Procedure).

The presentation level of the sound texture was fixed at 55 dB SPL and the noise level varied from trial to trial, with an SNR range of −18 dB and +12 dB (in 6 dB increments). Each stimulus was 2 s in duration, with an inter-stimulus-interval of 400 ms.

**Experiment 4a procedure**. The experiments were conducted in-lab. Participants completed twenty blocks of trials, with one block for each of the twenty sound textures used in the experiment. Each block contained 2 presentations per SNR value, for a total of 12 trials, presented in random order. The experiment consisted of 240 trials in total. The order of the blocks was randomized for each participant.

At the beginning of a block participants were played a 2 s exemplar of the target texture without noise, so that they knew what to listen for. On each trial, participants heard the interval composed of noise plus the texture and the interval composed of only noise, in random order. Participants judged whether the target texture was present in the first or second interval. Every stimulus interval was generated from a unique noise segment and texture exemplar (a unique synthetic exemplar or excerpt from the real-world recording, depending on the version of the experiment), such that the participants never heard the same waveform twice within or across trials. Participants had the option of refreshing their memory of the target texture by listening to it again in isolation at any point in the block.

**Experiment 4a participants**. 10 participants completed the experiment with real texture sounds (7 female, mean age = 20.7, s.d. = 1.8, 10 self-reported as musicians). A different group of 10 participants completed the experiment with synthetic texture sounds (8 female, mean age = 22.4, s.d. = 2.71, 10 self-reported as musicians).

**Experiment 4b stimuli**. The same twenty sound textures used in Experiment 4a were used in Experiment 4b. The stimulus consisted of sound textures interrupted with a noise segment (Fig. 5a —lower panel). The stimulus was constructed by concatenating a 2-s excerpt of sound texture, followed by a 2 s Gaussian masking noise, followed by an additional 1 s of the synthetic texture. The texture excerpts were the first 2 s and the last 1 s of a 5 s excerpt. Stimuli were constructed as in Experiment 1. We ran two versions of the experiment: one where the texture excerpts were taken from real-world recordings, and one where they were synthetic textures. For the real-world recordings, the texture excerpts were randomly selected 5 s excerpts from the 7 s source recording. For the synthetic textures, the stimuli were random excerpts sampled without replacement from twelve 5 s samples synthesized from the statistics of each real-world texture. The presentation level of the sound texture excerpts was fixed at 55 dB SPL. The SNR of the interrupting noise segment varied between −18 dB and +12 dB (in 6 dB increments).

The experiment also included twenty control trials, one for each of the twenty sound textures, in which the texture was physically continuous during the interrupting noise segment, with an SNR value of +6 dB.

**Experiment 4b procedure**. On each trial, participants heard a 5 s stimulus as described above. Participants judged whether the texture was continuous or discontinuous during the noise. The experiment consisted of 280 trials: 2

presentations of the 120 main experiment trials (6 masker levels × 20 textures) and 2 presentations of the 20 control trials. The order of the trials was randomized for each participant with the constraint that the same texture never occurred in consecutive trials.

**Experiment 4b participants**. The participants were those from Experiment 4a.

**Experiment 5 stimuli**. The same twenty textures from Experiment 4a/b were used in Experiment 5. The stimuli were constructed by interrupting a 5 s synthetic texture with 2 s of Gaussian noise or a combination of Gaussian noise and silence, the latter case resulting in a gap between the texture and the noise. The synthetic textures were random, drawn from a set of eight 5-s samples synthesized from the statistics of each real-world texture. A different synthetic exemplar was used on each trial. The experiment had four conditions distinguished by the presence or absence of gaps: (i) no gaps (the sound texture and noise were contiguous), (ii) a gap before and after the noise (iii) a gap after the noise (iv) a gap before the noise (Fig. 6a). The gaps were always 200 ms in length and replaced the beginning or end of the noise. The presentation level of the sound texture excerpt was set to 55 dB SPL. The level of the intermediate noise segment was set individually for each sound as that which would mask the inducer texture were they presented simultaneously (validated by Experiment 4a). This level varied somewhat from texture to texture, but was always between 67.5 and 75 dB SPL (corresponding to an SNR between −12.5 and −20 dB; see Supplementary Table 4).

In addition to trials with the main experimental conditions, we included 48 control trials in which the texture was physically present during the noise, to confirm that participants could perform the task. The control trials were constructed to mimic the response contours by modulating the amplitude of the synthetic texture during the intermediate noise segment (see schematics in Fig. 6c). The stimulus consisted of a 2 s inducer texture, followed by a 2 s segment of "contoured" texture plus noise, followed by 1 s of texture. For the time-varying contour shapes, the contoured texture was constructed by multiplying the texture by a raised cosine envelope that varied between 0 and 1 with a phase set to yield the desired shape. The SNR value between the unattenuated inducer texture and noise was +6 dB, such that the texture was audible when superimposed on the noise (compare to Fig. 2b, left panel). The portion of the synthetic texture corresponding to the response contour extended for 2 s and co-occurred with the noise segment. The six contour shapes were presented twice for each of the four experiment conditions, with the texture randomly selected from the 20 textures used in the main experiment. For the control conditions with a silent gap, the first and/or last 200 ms of the texture + noise combination was replaced with silence (with 20 ms onset and offset ramps), such that each contour was presented in each of the four stimulus configurations, to better assess whether participants could identify the contour shape across the various gap arrangements.

**Experiment 5 procedure**. The experiment was conducted in-lab. On each trial participants heard a 5 s stimulus and selected one of 6 response contours to indicate what they heard: continuous throughout, fade-out, present at beginning and end of the noise ('dip'), present in the middle of the noise ('glimpse'), fade-in, and absent throughout. Prior to the main experiment, participants performed 24 practice trials, which had the same form as the control trials, but with combinations of textures and conditions that were not used in the control trials. Participants received feedback for the practice trials.

The experiment consisted of 232 trials: two presentations of the 4 main conditions with each of the 20 textures, 48 control trials, and the initial 24 practice trials. The order of the practice trials was randomized but always occurred at the beginning of the experiment. The order of the main experiment trials and control trials was randomized.

**Experiment 5 participants**. The 10 participants from Experiment 4a and 4b (the version with synthetic sound textures) completed Experiment 5.

**Experiment 6 texture step and morph synthesis**. Synthetic texture stimuli were generated using a variant of the McDermott and Simoncelli[14] sound texture synthesis system. As introduced in a recent publication[20], the original texture synthesis system was modified to facilitate the generation of "texture morphs" (sound textures generated from statistics sampled at points along a line between two textures) and "texture steps" (sound textures that underwent a change in their statistics at some point during their duration).

We measured the statistics of 50 real-world texture recordings, averaged their statistics, and used these average statistics to define the "mean" texture. The statistics measured from individual real-world texture recordings defined a "reference" texture that varied across blocks. The cochlear envelope means ($\mu_k$) of the reference were set equal to those for the mean texture, such that spectral content was constant across stimuli (so that any integration effects would be likely to reflect higher-order statistics). We synthesized texture morphs by sampling discrete points along a line in the space of statistics between the mean and the reference. We generated texture steps so that the stimulus statistical properties changed at some point in time by stepping from one set of texture statistics to another. For the texture steps, we first synthesized a texture from one set of

statistics. We then synthesized from a second set of statistics from the line between the mean and reference, using the first synthetic texture as the seed to the synthesis system. The resulting synthetic textures were then windowed (with rectangular windows) and summed to yield a texture step with the desired change in statistics that occurred at a specific point in time. This procedure produced signals whose statistics changed over time without introducing artifactual discontinuities that might otherwise result from concatenating signals with different statistics.

**Experiment 6 stimuli**. Four reference textures were used (River running over shallows, Pneumatic drill, Applause, and Lawn edger). These textures were selected because they elicited a consistently high continuity response when interrupted with masking noise. Each trial presented a "step" stimulus followed by a "morph" stimulus.

The step stimulus statistics began at either 25% or 75% of the distance between the mean and reference, and stepped to the 50% point between the mean and reference. The step stimulus was 5 s in duration. There were four conditions, each with a different type of step stimulus, each of which had two step directions (beginning at either the 25% or 75% point; Fig. 7c). The first two conditions included a step positioned either 1 s or 3 s from the endpoint. The third condition replaced an intermediate segment of the 3 s step with 2 s of Gaussian masking noise, ending 1 s from the endpoint. The fourth condition was identical to the third condition, except that the first 200 ms of the masking noise was replaced by silence. In all conditions, the texture was not physically present during the interrupting noise. Five exemplars were synthesized for each step condition for each reference texture.

The second stimulus on a trial (the morph) was generated with statistics drawn from one of 10 positions between the mean and the reference (0, 0.25, 0.35, 0.4, 0.45 0.55, 0.6, 0.65, 0.75, and 1, on a linear scale, where the number denotes the fractional position between mean and reference, with 0 denoting the mean and 1 denoting the reference). The morph duration was two seconds. The step and morph were separated by an inter-stimulus-interval of 250 ms. The sound texture portion of the signal was presented at 55 dB SPL. When present, the masking noise was set between 14–18 dB above the level of the texture (Supplementary Table 5), with the exact level set individually for each texture as that necessary to mask the texture when the noise and texture were superimposed.

**Experiment 6 procedure**. The experiment was conducted in-lab. Trials were blocked by the reference texture. Each block presented one trial for each of the 10 morph positions paired with a randomly selected step condition. Each step condition occurred once with each morph position for each reference texture across the experiment. The order of the blocks was always random subject to the constraint that two blocks with the same reference texture never occurred consecutively. The stimulus used on a trial was randomly selected from five pre-generated synthetic textures for each condition and reference texture. At the start of each block participants were played a 5 s excerpt of the reference texture, which they could listen to as many times as they wished. Participants had the option of listening to the reference texture again between trials during the block.

Participants selected the interval that was most similar to the reference texture. Participants were informed that the step interval could change over time and could contain an interrupting noise segment. They were instructed to base their judgments on the endpoint of the step interval, a task that we validated in a previous publication[20]. Feedback was not provided.

Participants began by completing a practice session (on the same day) consisting of 4 blocks (each with a different randomly selected reference texture) of 6 trials to familiarize participants with the task and the reference sounds. In the practice session the first stimulus did not contain a step, and was instead generated with constant statistics drawn from the midpoint between the reference and mean, and the second stimulus was generated from statistics drawn from one of six positions (0 - mean, 0.25, 0.4, 0.6, 0.75, 1 - reference). Both stimuli were 2-s in duration. Feedback was given following each practice trial.

**Experiment 6 participants**. To restrict the analysis to participants who could perform the task (which was more difficult than the others used in this paper), we excluded participants who did not perform at least 85% correct when the morph was set to the statistics of the reference or mean (i.e., the most discriminable difference used in the experiment) for at least one of the step conditions. Sixteen participants completed the experiment, of which thirteen met the inclusion criterion (9 female, mean age = 21.2, s.d. = 1.5, 13 self-reported as musicians). The participants were unique to this experiment, and did not participate in any other experiments.

**Experiment 7 stimuli**. We selected 20 sound textures and 20 non-textures from a larger set as those that had high and low values of sound stationarity (Fig. 8a). Each stationary sound (texture) was paired with a variable sound (non-texture). The authors attempted to pair the sounds such that they could plausibly occur in natural listening environments (e.g. applause paired with music; see Supplementary Table 6).

Each trial began with a 2 s cue sound, followed by a 400 ms silent gap, followed by a 5 s stimulus. There were three conditions, differentiated by the cue sound: (i)

texture, (ii) non-texture, and (iii) superposition of texture and non-texture. The 5 s stimulus was the concatenation of 2 s of the superposition of texture and non-texture (the inducer), followed by 2 s of Gaussian noise (the masker), followed by another 1 s of the superposition of texture and non-texture. The sound segments were cross-faded over 20 ms with a raised cosine ramp. The waveform of the cue and initial 2 s of the texture, non-texture, or their superposition were identical. The noise level was set by the authors to plausibly mask the texture + non-texture inducer (Supplementary Table 6).

The experiment also included forty control trials to confirm task comprehension and compliance. Similar to the main experimental trials, the control trials cued participants to report the continuity of a sound: either the texture, the non-texture or the combined sound. But unlike the main experimental trials, the inducer in the control trials was higher in level than the noise and the cued sound was either physically present (continuous) or absent during the noise (Supplementary Fig. 8). The texture and non-texture in the inducer were equal in level, and the inducer (texture + non-texture) was 55 dB SPL. The noise was set to 6 dB below the cued sound (either the texture, non-texture or their combination, depending on the condition). The cue was the same level as the corresponding source signal(s) within the inducer. There were 5 continuous control conditions, where the cued sound was physically present during the noise, but where the un-cued sound was present or absent during the noise depending on the condition. There were also 5 discontinuous control conditions, where the cued sound was physically absent during the noise, but where the un-cued sound was present or absent during the noise depending on the condition. If listeners were reporting the subjective presence of the cued sound (as instructed), they should report continuity on the 5 continuous control conditions but not the 5 discontinuous control conditions. The results of the 5 continuous control conditions and the 5 discontinuous control conditions were analyzed together (Supplementary Fig. 8). There were a total of 200 possible control trials (10 conditions crossed with 20 inducer sound pairs).

**Experiment 7 procedure**. The experiment was conducted in-lab. On each trial, participants heard the cue sound followed by the stimulus. Participants judged whether the cued sound was continuous or discontinuous during the interrupting noise. Participants were instructed to respond with continuous only if the cued sound continued during the noise. For instance, if the cue sound was the sound pair, the listeners would respond with "continuous" if the sound pair continued during the noise and "discontinuous" if only one sound continued or neither sound continued during the noise. Participants performed 160 trials in total: 120 trials of the three main conditions and 40 control trials. Each of the three main conditions occurred twice for all twenty sounds. The control trials were randomly selected for each participant from the set of 200 possible stimulus/condition pairings, with the constraint that each inducer sound pair occurred twice and each condition occurred four times. The order of the trials was randomized for each participant.

Prior to the main experiment, participants performed 20 practice trials with feedback. The practice trials were identical in structure to the control trials, but with distinct stimulus/condition pairings. The feedback indicated whether the cued sound was in fact continuous or discontinuous during the noise.

**Experiment 7 participants**. 10 participants completed the experiment (5 female, mean age = 23.1, s.d. = 1.9, 9 self-reported as musicians). These participants were exclusive to Experiment 7.

**Sample size determination**. The sample size for each experiment was chosen to yield stable results based on split-half reliability of the results in pilot experiments.

**Sample sizes for experiment 1**. *In-lab*. We ran a pilot version of the experiment ($N = 8$). The sound set was slightly different from that of Experiment 1. We calculated the split-half reliability (Pearson's correlation coefficient) of the main result (proportion of continuous responses for each of the inducer sounds) for sample sizes ranging from $N = 2$ to 8 with random splits of the participants ($N = 10,000$). We found the split-half reliability increased with sample size from 0.60 ($N = 2$) to 0.87 ($N = 8$). We extrapolated that a sample size of 13 would have a likely test-retest correlation greater than 0.9, and chose this as our sample size.

*Online*. We used the power analysis for Experiment 2, which used a similar task and was also conducted online. This yielded a sample size of 80, which we slightly exceeded.

**Sample sizes for experiment 2**. We ran a pilot version of the experiment ($N = 44$). The experiment differed from Experiment 2 in two respects: (i) participants completed all sound/condition pairings, resulting in multiple trials with the same sounds, and (ii) the sound set was slightly different. We performed a power analysis by selecting subsets of 24 trials (to match the number of trials in Experiment 2). We measured the Pearson's correlation coefficient between random splits of participants for sample sizes ranging from $N = 2$ to 44. The split-half reliability increased with sample size from 0.17 ($N = 2$) to 0.84 ($N = 44$). We extrapolated that a sample size of 80 would give an expected split-half reliability greater than 0.9, and chose this as the sample size.

**Sample sizes for experiment 3**. We ran a pilot experiment ($N = 6$). The pilot experiment included 40 sounds (rather than the 20 sounds used in Experiment 3), and the noise extended for only 2 s after the inducer (rather than the 4 s used in Experiment 3). Split-half reliability increased with sample size from 0.47 ($N = 2$) to 0.83 ($N = 6$). We chose a sample size of 10 to yield an expected reliability in excess of 0.9.

**Sample sizes for experiment 4a**. We ran a pilot version of the experiment ($N = 9$) with real-world sound textures that used a slightly different set of textures and a reduced range of SNR conditions (spanning −18 dB to +6 dB in 6 dB increments). We calculated the split-half reliability (Pearson's correlation coefficient) of the main result (proportion correct for each of the SNR conditions) for sample sizes ranging from $N = 2$ to 8 with random splits of the participants ($N = 10,000$). The mean split-half reliability increased with sample size from 0.86 ($N = 2$) to 0.96 ($N = 8$). We selected a target sample size of 10 as that which would likely yield a split-half correlation of at least 0.9.

**Sample sizes for experiment 4b**. We ran an analogous pilot version of Experiment 4b. Using the same power analysis procedure described for Experiment 4a, we found the split-half reliability increased with sample size from 0.71 ($N = 2$) to 0.91 ($N = 8$). We selected a target sample size of 10 as that which would provide an expected reliability correlation greater than 0.9.

**Sample sizes for experiment 5**. We ran a pilot version of the experiment ($N = 8$). The experiment was identical to Experiment 5 except that the inducer sounds were real-world texture recordings and the sound set was slightly different. We calculated the split-half reliability (Pearson's correlation coefficient) of the mean results (the proportion responses for each contour shape) for sample sizes ranging from $N = 2$ to 8 with random splits of the participants ($N = 10,000$). The split-half reliability increased with sample size from 0.73 ($N = 2$) to 0.91 ($N = 8$). We selected a sample size of 10 as that which would provide an expected test-retest correlation greater than 0.9.

**Sample sizes for experiment 6**. We had performed power analyses for this type of experiment in a previous publication[20]. We chose a sample size of 10 as this yielded an expected split-half reliability of the mean results that exceeded 0.9.

**Sample sizes for experiment 7**. The methodology of this experiment was similar to that of the in-lab version of Experiment 1, and we targeted the same sample size identified for that experiment ($N = 10$) for an expected reliability in excess of 0.9.

**Normality**. Data was assumed to be normally distributed and was evaluated as such by eye.

**Statistics and data analysis for experiment 1**. We evaluated the reliability of the online data by computing Pearson's correlation coefficient between the mean proportion of continuous responses for splits of participants (using 10,000 random splits). The reported correlation is the mean over the resulting 10,000 correlation values. Continuity judgments were compared to inducer stationarity using Pearson's correlation coefficient.

**Statistics and data analysis for experiment 2**. The effects of sound category and noise duration were assessed using a multi-level repeated-measures analysis of variance (ANOVA). Paired comparisons were made with two-tailed t-tests.

**Statistics and data analysis for experiment 3**. A repeated-measures ANOVA was used to evaluate the variation in the temporal extent of continuity across inducer sounds. The extent of perceived continuity was compared to inducer stationarity using Pearson's correlation coefficient.

**Statistics and data analysis for experiment 4a/b**. Repeated-measures ANOVAs were used to test for effects of SNR on masking and continuity judgments.

To compare the threshold SNRs that produced masking and illusory continuity (as shown in Supplementary Fig. 4), we fit logistic functions to the data from each experiment. Using these fits, we defined the masking threshold as the SNR producing a proportion correct of 0.833 in the texture masking experiment (4a) and the continuity threshold as the SNR producing judgments of continuity on 0.667 of the trials for texture continuity experiment (4b). These values represent 2/3 of the dynamic range of the psychometric functions (the masking curve spans 0.5 and 1, whereas the continuity curve spans 0 and 1). We compared the threshold SNR values for masking and illusory continuity with a Spearman's rank correlation coefficient across sounds (Supplementary Fig. 4c).

**Statistics and data analysis for experiment 5**. Single sample or paired two-tailed t-tests were used to compare results for individual conditions to chance, and for pairwise comparisons between conditions, respectively.

**Statistics and data analysis for experiment 6**. We fit psychometric (logistic) functions to the mean results for each step condition and step direction (i.e., the curve plotting the proportion of trials where the morph was judged to be more similar to the reference, for each morph position), and from the fitted functions obtained the point of subjective equality for each step condition and direction. We quantified the step-induced bias as the difference between the points of subjective equality for the two step directions for each condition. Confidence intervals on these bias measurements were derived by bootstrap (10,000 samples)[20]. Statistical significance of differences in the bias between conditions was estimated from the bootstrap distributions of the bias by fitting a Gaussian to the bootstrap distribution and then computing the p-value from the Gaussian.

**Statistics and data analysis for experiment 7**. Differences between conditions were evaluated with paired two-tailed t-tests. An unpaired, two-tailed t-test was used to evaluate differences in continuity for the same two sound categories (textures and non-textures) between Experiment 7 and Experiment 1.

**Reporting summary**. Further information on research design is available in the Nature Research Reporting Summary linked to this article.

## Data availability

The source data underlying Figs. 2c, d, f–h, 3c, 4c–e, 5b, 6c, d, 7d, and 8d and Supplementary Figs. 1c–g, 2, 3b, c, 4, 5c, d, 7, 8b are provided as a Source Data file. All other data are available upon request.

## Code availability

All code is available upon request.

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

## Acknowledgements

The authors thank the McDermott laboratory for comments on the manuscript, in particular M. Cusimano and J. Feather for comments on the penultimate draft, and B. Anderson for helpful discussions. This work was supported by a McDonnell Foundation Scholar Award to J.H.M. and NIH Grant No. R01DC014739. The funding agencies were not otherwise involved in the research, and any opinions, findings and conclusions or recommendations expressed in this material are those of the authors and do not necessarily reflect the views of the McDonnell Foundation, or NIH.

## Author contributions

R.M. and J.H.M. designed the experiments. R.M. implemented experiments, and collected and analyzed the results. R.M. and J.H.M. wrote the paper.

## Competing interests

The authors declare no competing interests.
