## [Peer Review File · Nature Communications]

Reviewers' comments:

Reviewer #1 (Remarks to the Author):

In their manuscript entitled "Illusory sound texture reveals multi-second statistical completion in auditory scene analysis", McWalter & McDermott provide compelling evidence for the neural representation of sound statistics through a series of experiments, which trace (but also extend) the principles of previous research into illusory sounds. A particularly interesting finding in my opinion is the differentiation between predictability and statistical representation (Exp.2). Overall the manuscript is quite mature and well written, leaving only a few points to be addressed. I see two major points for clarification, the first one requiring an additional experiment and some additional analysis.

Major points:

1. The authors convincingly show that the sounds that they identify as statistically stationary (i.e. (a subset of) auditory textures, see below) are particularly likely to be perceived in the context of intermittent, masking noise. However, they focus only on this characterization, while not exploring alternative hypotheses, e.g. are the statistically stationary sound maybe just closer in their spectral appearance to the noise, and hence, they are more like to be confused with the noise itself? At this point it should be also emphasized that the noise itself falls into the class of statistically stationary, and it is hence not impossible that similarity is underlying the illusion, rather than the particular properties of the auditory textures. In order to address this point, I would like the authors perform three independent tests:

a) quantify the difference (in the measured statistics) between all tested sounds to the noise, and verify (if true) that the statistically stationary ones are not 'just' more similar to the noise.

b) as a basic alternative explanation quantify the spectrotemporal density of each stimulus and again evaluate the difference to the noise as a potential predictor for the duration of the illusory continuation.

c) run a separate control experiment (e.g. for experiment 1, on Mechanical Turk), where the masking noise is based on the measured statistics more similar to another group of stimuli, and test whether still the statistically stationary sounds are the ones that are dominantly heard as an illusory continuation.

2. While I generally agree with the notion of stationarity in statistics as a good descriptor for auditory textures, I think it should be clarified early on, what this precisely means, since in this case it refers to the (well-chosen, but still somewhat arbitrary) statistics from the texture synthesis model, which could be characterized as neurally-inspired, but also of low complexity. Probably most (if not all) sounds in the sample can be described with statistics of some order, and I think a key point to make here is that the stationary statistics are of low degree/complexity, which then together makes a good definition for auditory textures, in my opinion.

Minor:

- Out of curiosity: the 'coin rolling on plate' appears to me to be a very basic stimulus, with constant statistics and high degree of predictability. Why does it score so low in your stationarity metric? Does it change in speed, or fall over?

- In Fig.2, something goes wrong after Fig.2d, which is not correctly mentioned in the text (or not at all?) and the subsequent panels seem to be shifted by one letter. Please check, and describe the control condition more properly.

Reviewer #2 (Remarks to the Author):

This well-written manuscript presents a series of seven experiments documenting a new continuity

illusion based on statistical regularity of sound texture. To the best of my knowledge, the findings presented in this manuscript are very novel and should have a significant impact on research spanning from hearing sciences, audiology, psychology and neuroscience. I have some minor suggestions.

The manuscript is somewhat dense and fairly long. One wonders whether all seven experiments are really needed to make the point. Also some materials seem more appropriate for a supplementary section than the core manuscript. For instance, the power calculation for each experiment could be moved to the section on supplementary material.

What are the possible mechanisms? Perhaps attention plays a role? I think the manuscript would benefit from having a discussion on the possible mechanisms underlying the illusion.

I would suggest adding more details about participant recruitment. Were the participants screened for hearing loss, and/or music training? Also, were there any participants excluded because they did not perceive the illusion?

In the abstract, the conclusion is analogous to what was proposed for the continuity illusion. So one wonders what is really novel here. How do their findings advance current views and models of auditory scene analysis in general and the continuity illusion in particular?

Line 58, I would suggest replacing "providing a background to an auditory scene" with "providing a background to other transient auditory events within a complex auditory scene"

Line 63-64, this sentence is awkward. I am not sure what the authors mean by "filled in" statistical representations." It seems misleading and/or oversimplified of the actual description of the continuity illusion. Along the same line, the manuscript would benefit from a brief description of current models or explanations of the continuity illusion and how the current study helps advance these models and theories.

Line 118, I would suggest replacing "was interrupted with a 2 s" with "was interrupted and the 2 s gap was filled with white masking noise."

Line 299, replace "signal-to-noise" with "SNR"

Line 384-385, I would encourage the authors to expand on the idea of "retrospective filling in." There is increasing evidence that context provided after a word in noise can facilitate its identification (Chan & Alain, 2019; Golestani et al. 2009; 2013). Discussing their findings in light of these studies showing the importance of reflective attention in auditory scene analysis would broaden the appeal of their study to a larger audience.

Line 683, please add a citation.

Line 802, replace "trails" with "trials"

Line 848, replace "control trials" with "control"

Line 868 and 932, please correct reference format [Woods et al., 2017].

Line 872-874 and elsewhere, please clarify what is a correct response (see also section 932-943). In the present study, this is a bit confusing because participants are required to make subjective responses on each trial. I understand that there are catch trials in which the inducer was present throughout the noise. So, does the accuracy score take catch trials into account? Also, it might be more appropriate to refer to a likelihood of hearing the illusion rather than an actual objective measure it. That is, for example, "participants were more likely to report hearing the inducer when

..."

Line 1001, please fit the citation.

Line 1196, please clarify whether the sessions were performed on the same day or different day.

McWalter & McDermott, Response to Reviews

Reviewer #1 (Remarks to the Author):

In their manuscript entitled "Illusory sound texture reveals multi-second statistical completion in auditory scene analysis", McWalter & McDermott provide compelling evidence for the neural representation of sound statistics through a series of experiments, which trace (but also extend) the principles of previous research into illusory sounds. A particularly interesting finding in my opinion is the differentiation between predictability and statistical representation (Exp.2). Overall the manuscript is quite mature and well written, leaving only a few points to be addressed. I see two major points for clarification, the first one requiring an additional experiment and some additional analysis.

Thank you.

Major points: 1. The authors convincingly show that the sounds that they identify as statistically stationary (i.e. (a subset of) auditory textures, see below) are particularly likely to be perceived in the context of intermittent, masking noise. However, they focus only on this characterization, while not exploring alternative hypotheses, e.g. are the statistically stationary sound maybe just closer in their spectral appearance to the noise, and hence, they are more like to be confused with the noise itself? At this point it should be also emphasized that the noise itself falls into the class of statistically stationary, and it is hence not impossible that similarity is underlying the illusion, rather than the particular properties of the auditory textures.

We agree this is an alternative worth addressing. As described below, we conducted the requested experiment and analyses to rule it out. The results are fairly decisive, and we have added them to the manuscript.

In order to address this point, I would like the authors perform three independent tests:

a) quantify the difference (in the measured statistics) between all tested sounds to the noise, and verify (if true) that the statistically stationary ones are not 'just' more similar to the noise.

This analysis shows (unsurprisingly to us texture aficionados) that statistical distance to white noise is in fact correlated with stationarity ($r = 0.79$). As a result, the statistical distance to noise is also predictive of the continuity percept, albeit somewhat less so than stationarity ($r = 0.73$ vs. $r = 0.84$, $p < .001$ via bootstrap). Even though the similarity to noise does not predict the illusion as well as the stationarity measure, this analysis provides motivation for an experiment to more clearly distinguish the two possible root causes of the variation in the illusion across sounds.

This analysis is now shown in the new Supplemental Figure 1 and mentioned in the main text to motivate the additional experiment (described below).

b) as a basic alternative explanation quantify the spectrotemporal density of each stimulus and again evaluate the difference to the noise as a potential predictor for the duration of the illusory continuation.

We formulated measures of spectral and temporal density as the negative logarithm of the standard deviation of the cochleagram across either time and frequency. Each of these measures is high for white noise, as expected, but only the temporal density measure is strongly correlated with our stationarity measure (temporal: $r=.89$; spectral: $r = -0.28$), and only the temporal density measure was strongly predictive of perceptual continuity (temporal: $r=0.76$, $p<.001$; spectral: $r=-.22$, $p=.05$), albeit less so than our stationarity measure, which measures variation over time in several classes of statistic ($r=0.84$, significantly greater than the correlation with density; $p<0.001$). This underscores that it is homogeneity across time, as captured by our stationarity measure, that is most closely related to the illusion.

We also looked at a spectrotemporal density measure (the negative logarithm of the standard deviation across both time and frequency), which as expected was somewhat predictive of continuity ($r=0.32$), but less so than the temporal density measure, or our stationarity metric ($r=0.84$, significantly different from the correlation with density; $p<0.001$ via bootstrap). We opted to include the separate spectral and temporal density measures in the paper as they provide insight into the importance of stationarity vs. other noise-like properties that a signal can have.

We now describe this result in the main text and in the new Supplemental Figure 2, and define the measure in the Methods section.

From the main text:

“To further distinguish stationarity from similarity to white noise we separately examined the effect of uniformity across time from that of uniformity across frequency. To obtain comparable measures in the two dimensions we simply measured the variability of the cochleagram across time (averaged over frequency) or across frequency (averaged over time), yielding measures of temporal and spectral “density” for each inducer sound (Supplementary Fig. 2). Unsurprisingly, the temporal density of the inducer was correlated with our stationarity measure ($r=.89$) and predictive of illusory continuity ($r=.76$, $p<.001$). By contrast, the spectral density was marginally negatively correlated with illusory

continuity ($r=-.22$, $p=.05$). This analysis provides further evidence that stationarity, rather than similarity to noise, is critical to the illusion. “ (lines 208-218)

The new Supplemental Figure 2:

The description of the measures from the Methods:

“Spectral and Temporal Density Metrics

The spectral and temporal sound density was quantified as the negative logarithm of the variance along either axis of the cochleagram of a sound. We computed the cochleagram using the first two stages of the auditory texture model described above. For the temporal density we measured the variance of each frequency channel of the cochleagram across time, averaging over frequency to obtain a single measure. For the spectral density we measured the variance across frequency at each time point, averaging over time to obtain a single measure. Our analyses used the negative logarithm of these measures.” (lines 836-844)

c) run a separate control experiment (e.g. for experiment 1, on Mechanical Turk), where the masking noise is based on the measured statistics more similar to another group of stimuli, and test whether still the statistically stationary sounds are the ones that are dominantly heard as an illusory continuation.

Because the masker has to be able to mask the inducer sounds, and thus not contain any substantial gaps in the cochleagram, there are some limits to what is possible. We converged on a control version of Experiment 1 in which the masking noise was generated from the mean statistics of the set of 80 sounds from Experiment 1. As a result, the masking noise had intermediate stationarity, and the most stationary sounds were no longer most similar to the masking noise (unlike for Experiment 1, where the most stationary sounds also tended to be the most similar to the masker). Stationarity was thus dissociated from statistical distance to the masker ($r = -0.11$):

The results with this alternative masker are clear: stationarity remains predictive of whether people hear illusory continuity ($r = .78$, $p < .001$), whereas statistical distance to the masker is only weakly related ($r = -0.25$, $p = .03$):

Although there is some effect of the masker sound (reported continuity is overall a bit higher for the mean masker than for the white noise masker), the illusion is generally quite similar across the masker sounds. The demos from this new experiment are also compelling and have been added to our demo page.

We believe this experiment shows that stationarity is indeed the main factor determining whether sounds are heard to continue over long durations when masked, and that similarity to the masker does not matter much. We have added this experiment to the paper as the new Experiment 1b. The key results graph has been added to Figure 2, and a more expansive summary of the results has been added as the new Supplemental Figure 1. Thank you for providing the impetus to run the experiment.

Here is the addition to the main text:

“Because the masker used in the experiment was white noise, which itself is highly stationary, the results could in principle have been driven by the statistical similarity between the inducer sound and the masker (which is greater for more stationary inducer sounds; Supplementary Fig. 1) rather than by the stationarity of the inducer per se. We note that the masker in this experiment was constrained to be able to continuously mask each of the 80 inducer sounds, preventing us from

using highly non-stationary sounds as the masker (because they generally contain spectrotemporal gaps through which other signals can be glimpsed). However, it is possible to deviate from white noise to some extent. To assess the importance of masker-inducer similarity, we conducted an additional experiment (Experiment 1b) in which half the trials featured masking noise synthesized from the average statistics of the set of 80 inducer sounds (using texture synthesis; Fig. 2h, Supplementary Fig. 1). This “mean noise” masker was less stationary than white noise, and the statistical similarity between inducer and masker was dissociated from the inducer stationarity (Supplementary Fig. 1e).

Although there was some effect of the masker sound on reported continuity (listeners were more likely to report hearing the inducer sound as present with the mean masker than the white noise masker; Supplementary Fig. 1c), inducer stationarity remained strongly predictive of perceived continuity ($r = 0.78$, $p < .001$, with the less stationary masker). By contrast, statistical similarity to the mean noise masker was only weakly correlated with the illusion ($r = -0.25$, $p = .03$; Supplementary Fig. 1d). This result provides support for the notion that the stationarity of the inducer is critical for multi-second illusory continuity.”
(lines 183-206)

2. While I generally agree with the notion of stationarity in statistics as a good descriptor for auditory textures, I think it should be clarified early on, what this precisely means, since in this case it refers to the (well-chosen, but still somewhat arbitrary) statistics from the texture synthesis model, which could be characterized as neurally-inspired, but also of low complexity. Probably most (if not all) sounds in the sample can be described with statistics of some order, and I think a key point to make here is that the stationary statistics are of low degree/complexity, which then together makes a good definition for auditory textures, in my opinion.

We agree. We have clarified this in the text:

“Texture completion seemed potentially interesting in part because textures are believed to be represented with relatively low-order statistics that summarize acoustic information over time¹⁴⁻²¹ ...” (lines 62-65)

“Because textures are dense, stochastic, and in many cases defined only by relatively simple summary statistics, the perceptual completion appears to be mediated by extrapolated statistics.” (lines 90-93)

Minor:

- Out of curiosity: the 'coin rolling on plate' appears to me to be a very basic stimulus, with constant statistics and high degree of predictability. Why does it score so low in

your stationarity metric? Does it change in speed, or fall over?

This sound is non-stationary because the coin settles over the course of the recording, such that the rolling sound evolves over time. Upon publication we will post all the sounds used in these experiments so that readers can hear the sounds for themselves and use them if interested.

- In Fig.2, something goes wrong after Fig.2d, which is not correctly mentioned in the text (or not at all?) and the subsequent panels seem to be shifted by one letter. Please check, and describe the control condition more properly.

We have fixed these errors, which were due to a late addition to the figure.

Reviewer #2 (Remarks to the Author):

This well-written manuscript presents a series of seven experiments documenting a new continuity illusion based on statistical regularity of sound texture. To the best of my knowledge, the findings presented in this manuscript are very novel and should have a significant impact on research spanning from hearing sciences, audiology, psychology and neuroscience. I have some minor suggestions.

Thank you.

The manuscript is somewhat dense and fairly long. One wonders whether all seven experiments are really needed to make the point. Also some materials seem more appropriate for a supplementary section than the core manuscript. For instance, the power calculation for each experiment could be moved to the section on supplementary material.

We wanted to make the paper as definitive as possible, and so tried to be thorough in our choice of experiments. We think each experiment helps to flesh out the phenomenon in an important way. Given that the phenomenon is subjective it seemed important to attack it from a few different angles. But based on this comment we opted to put most of the results from the new experiment and analyses (conducted in response to the first reviewer) into the supplement (as the new Supplemental Figures 1 and 2).

We decided to maintain a single coherent methods section rather than put part of it into a supplementary document, as the sub-headings should allow the reader to skip sections that are not essential for their purposes. To keep the manuscript as short as possible we have responded to the reviewer comments with relatively concise text revisions.

What are the possible mechanisms? Perhaps attention plays a role? I think the manuscript would benefit from having a discussion on the possible mechanisms underlying the illusion.

We have added a paragraph to the discussion about the possible role of attention, which we think our experiments leave open:

“Our experiments leave open the role of directed attention in illusory texture. The traditional continuity illusion is believed to be robust to inattention⁴¹, like other aspects of auditory scene analysis that occur on relatively short time scales⁴²⁻⁴⁵. However, auditory “streaming” phenomena that evolve over seconds are often altered by attention^{46,47}, raising the possibility that multi-second illusory texture continuity might be similarly affected. The temporal extent of the illusion should enable experiments exploring such effects.” (lines 687-693)

I would suggest adding more details about participant recruitment. Were the participants screened for hearing loss, and/or music training? Also, were there any participants excluded because they did not perceive the illusion?

We have added detail to this section of the Methods. Participants all had self-reported normal hearing. We recorded whether participants self-reported musical training (now added to the description of each experiment in the Methods), but did not screen for this.

No participants were excluded based on whether they heard the illusion. We excluded participants only for non-compliance with task instructions. Participants in the in-lab experiments were generally compliant (evidenced by good performance on the control trials, as shown in the paper). It was only necessary to exclude participants for Experiment 6, which required subtle discriminations of textures.

Online participants had to pass our “headphone check” test (Woods et al., 2017), and then completed a practice experiment including the control conditions where the inducer stimulus was physically present or absent (and where the inducer was higher in level than the masker, so that continuity was only heard if the inducer was physically present). Participants had to give the correct answer on these trials in order to proceed to the main experiment.

The above information was included in the original manuscript, but we have made it easier to find by placing it in the initial general sections on Experimental Procedures, where we now explicitly state that no one was excluded for not hearing the illusion. Our sense is that the illusion works on pretty much everyone.

Excerpt from updated Experimental Procedure for In-Lab Experiments:

**“Participants were not screened for inclusion based on whether they heard the illusion or not. The only experiment where participants were excluded was Experiment 7, which involved a subtle discrimination tasks, and where we excluded participants who could not perform this task above a criterion level.”
(lines 854-858)**

Excerpt from updated Experimental Procedure for Online Experiments:

**“Finally, to eliminate participants who were not fully focused on the task, we included practice trials at the beginning of the experiment (described in more detail below). On these trials the inducer stimulus was was physically and audibly either present or absent during the noise (because it was higher in level than the noise), such that the stimulus was outside the regime of illusory continuity and there was an unambiguous correct answer. Performance on these trials thus provided a measure of task compliance independent of whether a participant heard the illusion or not. We excluded participants who performed at less than 75% correct on these practice trials. As with the in-lab experiments, participants were not screened for inclusion based on whether they heard the illusion or not.”
(lines 876-885)**

In the abstract, the conclusion is analogous to what was proposed for the continuity illusion. So one wonders what is really novel here. How do their findings advance current views and models of auditory scene analysis in general and the continuity illusion in particular?

We are obviously limited by space, but have modified the last sentence of the abstract to emphasize that the illusion implicates statistical representations that operate over long time scales, which is a key novel aspect to the illusion:

“The illusion appears to represent an inference about whether the background is likely to continue during concurrent sounds, providing a stable statistical representation of the ongoing environment despite unstable sensory evidence.”

Line 58, I would suggest replacing “providing a background to an auditory scene” with “providing a background to other transient auditory events within a complex auditory scene”

Done.

Line 63-64, this sentence is awkward. I am not sure what the authors mean by “filled in”

statistical representations.” It seems misleading and/or oversimplified of the actual description of the continuity illusion.

We have changed the phrasing of this sentence to explicitly state the hypothesis that statistical representations might be extrapolated over time:

“...raising the question of whether statistical representations could be extrapolated over time” (lines 64-65)

Along the same line, the manuscript would benefit from a brief description of current models or explanations of the continuity illusion and how the current study helps advance these models and theories.

We included several discussion paragraphs citing and reviewing previous psychophysical and neurophysiological results on the continuity illusion in light of our findings. These now emphasize the bottom line that models of auditory scene analysis need to work with statistical representations, and that our results provide evidence for explicit representations of inferred sound.

Here are the relevant sections of the discussion:

“A second difference with classical continuity effects is that the filling in of texture is statistical. Unlike tones or speech, textures are stochastic, and on the time scales at which the continuity occurs can only be described and extrapolated in terms of the distribution of their features. There are some previously documented examples of perceptual completion of abstracted properties of sound, in that frequency and amplitude modulation appear to be filled in at a level of representation that discards phase^{28,29}. Texture completion appears to be a more extreme version of this sort of phenomenon, in that the content that is completed reflects time-averaged properties of the inducing stimulus. The results underscore the need for models of auditory scene analysis to involve statistical representations.” (lines 634-643)

“Experiments with traditional continuity illusions have in some cases suggested explicitly “filled in” neural representations, envisioned as persistent activity from the inducer stimulus continuing during the noise. Evidence for persistent activity has been seen in primary auditory cortex of cats and monkeys for the original tone continuity illusion^{30,31}. And several studies in humans have reported evidence of explicit filling in during phonemic restoration³²⁻³⁴. On the other hand, other studies of tone continuity have suggested more implicit representations of the continuity, with the tone onsets and offsets being suppressed by the noise³⁵⁻³⁷, and one report that illusory tones do not produce aftereffects comparable to those induced by real tones³⁸.

Our results are suggestive of an explicit representation of illusory texture. We found evidence that illusory texture biases the perception of subsequent texture, as though the illusory texture was incorporated into the integration window for estimating texture statistics (Experiment 6). This result is consistent with the idea that perceptual continuity is mediated by persistent activity at the representational stage that is averaged to yield texture statistics. Illusory texture could thus involve a conceptually similar mechanism to that proposed for tones, though in a higher-order representation (e.g. modulation filters^{39,40}), and with persistent activity lasting much longer than with tones. But given that the continuity appears to be driven by the statistical properties of a sound, and that these statistics are derived via a multi-second integration process²⁰, any persistent activity may be driven by feedback from a representation of those statistics, the neural locus of which remains unclear. The strength and duration of the illusion should enable neurophysiological experiments to more definitively probe its neural basis.” (lines 661-685)

Line 118, I would suggest replacing “was interrupted with a 2 s” with “was interrupted and the 2 s gap was filled with white masking noise.”

We have rephrased the sentence in question, which now reads “each sound was interrupted, with a 2 s segment of the sound replaced with white masking noise”.

Line 299, replace “signal-to-noise” with “SNR”

Done.

Line 384-385, I would encourage the authors to expand on the idea of “retrospective filling in.” There is increasing evidence that context provided after a word in noise can facilitate its identification (Chan & Alain, 2019; Golestani et al. 2009; 2013). Discussing their findings in light of these studies showing the importance of reflective attention in auditory scene analysis would broaden the appeal of their study to a larger audience.

We agree that these studies are conceptually related and interesting. Due to space constraints we are unable to discuss them at length, but have added references to the cited studies.

Line 683, please add a citation.

Done.

Line 802, replace “trails” with “trials”

Done.

Line 848, replace “control trials” with “control”

Done.

Line 868 and 932, please correct reference format [Woods et al., 2017].

Done.

Line 872-874 and elsewhere, please clarify what is a correct response (see also section 932-943). In the present study, this is a bit confusing because participants are required to make subjective responses on each trial. I understand that there are catch trials in which the inducer was present throughout the noise. So, does the accuracy score take catch trials into account? Also, it might be more appropriate to refer to a likelihood of hearing the illusion rather than an actual objective measure it. That is, for example, “participants were more likely to report hearing the inducer when ...”

We have clarified here and elsewhere that the practice trials, like the control trials, contained stimuli where the inducer sound was higher in level than the masker and was physically and audibly present or absent during the masker, such that there was an unambiguous correct answer. These trials thus do not measure the illusion, but rather measure task comprehension/compliance.

This section now reads:

“As described above, on these trials the inducer sound was higher in level than the masker, and was physically and audibly present or absent during the noise, such that there was an unambiguous correct answer. If the participant’s performance met or exceeded 75% correct on the practice section, they continued to the main experiment, otherwise the session was ended.” (lines 987-991)

Line 1001, please fit the citation.

Done.

Line 1196, please clarify whether the sessions were performed on the same day or different day.

Done. The text now reads:

“Participants began by completing a practice session (on the same day)...” (line 1316)

REVIEWERS' COMMENTS:

Reviewer #1 (Remarks to the Author):

The authors have masterfully addressed my main concerns of spectrotemporal similarity in the new experiments (SF.1/2). Thanks for taking the time and effort for fully addressing them!

Minor:

- Typo in the title of Sup. Fig.2a/b: denstiy

Reviewer #2 (Remarks to the Author):

The authors have done an excellent job at revising their manuscript. I am very happy with their responses and I do not have additional comments nor suggestions.

REVIEWERS' COMMENTS:

Reviewer #1 (Remarks to the Author):

The authors have masterfully addressed my main concerns of spectrotemporal similarity in the new experiments (SF.1/2). Thanks for taking the time and effort for fully addressing them!

Thank you.

Minor:

- Typo in the title of Sup. Fig.2a/b: denstiy Fixed.

Reviewer #2 (Remarks to the Author):

The authors have done an excellent job at revising their manuscript. I am very happy with their responses and I do not have additional comments nor suggestions.

Thank you.